# Ongoing, rational calibration of reward-driven perceptual biases

**Yunshu Fan, Joshua I Gold, Long Ding***

Neuroscience Graduate Group, Department of Neuroscience, University of Pennsylvania, Philadelphia, United States

**Abstract** Decision-making is often interpreted in terms of normative computations that maximize a particular reward function for stable, average behaviors. Aberrations from the reward-maximizing solutions, either across subjects or across different sessions for the same subject, are often interpreted as reflecting poor learning or physical limitations. Here we show that such aberrations may instead reflect the involvement of additional satisficing and heuristic principles. For an asymmetric-reward perceptual decision-making task, three monkeys produced adaptive biases in response to changes in reward asymmetries and perceptual sensitivity. Their choices and response times were consistent with a normative accumulate-to-bound process. However, their context-dependent adjustments to this process deviated slightly but systematically from the reward-maximizing solutions. These adjustments were instead consistent with a rational process to find satisficing solutions based on the gradient of each monkey's reward-rate function. These results suggest new dimensions for assessing the rational and idiosyncratic aspects of flexible decision-making.

*For correspondence:
lding@pennmedicine.upenn.edu

Competing interest: See
page 21

Reviewing editor: Peter
Latham, University College
London, United Kingdom

## Introduction

Normative theory has played an important role in our understanding of how the brain forms decisions. For example, many perceptual, memory, and reward-based decisions show inherent trade-offs between speed and accuracy. These trade-offs are parsimoniously captured by a class of sequential-sampling models, such as the drift-diffusion model (DDM), that are based on the accumulation of noisy evidence over time to a pre-defined threshold value, or bound (*Ratcliff, 1978*; *Gold and Shadlen, 2002*; *Bogacz et al., 2006*; *Krajbich et al., 2010*). These models have close ties to statistical decision theory, particularly the sequential probability ratio test that can, under certain assumptions, maximize expected accuracy for a given number of samples or minimize the number of samples needed for a given level of accuracy (*Barnard, 1946*; *Wald, 1947*; *Wald and Wolfowitz, 1948*). However, even when these models provide good descriptions of the average behavior of groups of subjects, they may not capture the substantial variability under different conditions and/or across individual subjects. The goal of this study was to better understand the principles that govern this variability and how these principles relate to normative theory.

We focused on a perceptual decision-making task with asymmetric rewards. For this task, both human and animal subjects tend to make decisions that are biased towards the percept associated with the larger payoff (e.g., *Maddox and Bohil, 1998*; *Voss et al., 2004*; *Diederich and Busemeyer, 2006*; *Liston and Stone, 2008*; *Serences, 2008*; *Feng et al., 2009*; *Simen et al., 2009*; *Nomoto et al., 2010*; *Summerfield and Koechlin, 2010*; *Teichert and Ferrera, 2010*; *Gao et al., 2011*; *Leite and Ratcliff, 2011*; *Mulder et al., 2012*; *Wang et al., 2013*; *White and Poldrack, 2014*). These biases are roughly consistent with a rational strategy to maximize a particular reward function that depends on both the speed and accuracy of the decision process, such as the reward rate per trial or per unit time (*Gold and Shadlen, 2002*; *Bogacz et al., 2006*). This strategy can be accomplished via context-dependent adjustments in a DDM-like decision process along two primary

dimensions (**Figure 1A**): (1) the momentary sensory evidence, via the drift rate; and (2) the decision rule, via the relative bound heights that govern how much evidence is needed for each alternative (**Ratcliff, 1985**). Subjects tend to make adjustments along one or both of these dimensions to produce overall biases that are consistent with normative theory, but with substantial individual variability (**Voss et al., 2004**; **Cicmil et al., 2015**; **Bogacz et al., 2006**; **Simen et al., 2009**; **Summerfield and Koechlin, 2010**; **Leite and Ratcliff, 2011**; **Mulder et al., 2012**; **Goldfarb et al., 2014**).

To better understand the principles that govern these kinds of idiosyncratic behavioral patterns, we trained three monkeys to perform a response-time (RT), asymmetric-reward decision task with mixed perceptual uncertainty (**Figure 1B**). Like human subjects, the monkeys showed robust decision biases toward the large-reward option. These biases were sensitive not just to the reward asymmetry, as has been shown previously, but also to experience-dependent changes in perceptual sensitivity. These biases were consistent with adjustments to both the momentary evidence and decision rule in the DDM. However, these two adjustments favored the large- and small-reward choice, respectively, leading to nearly, but not exactly, maximal reward rates. We accounted for these adjustments in terms of a satisficing, gradient-based learning model that calibrated biases to balance the relative influence of perceptual and reward-based information on the decision process. Together, the results imply complementary roles of normative and heuristic principles to understand how the brain combines uncertain sensory input and internal preferences to form decisions that can vary considerably across individuals and task conditions.

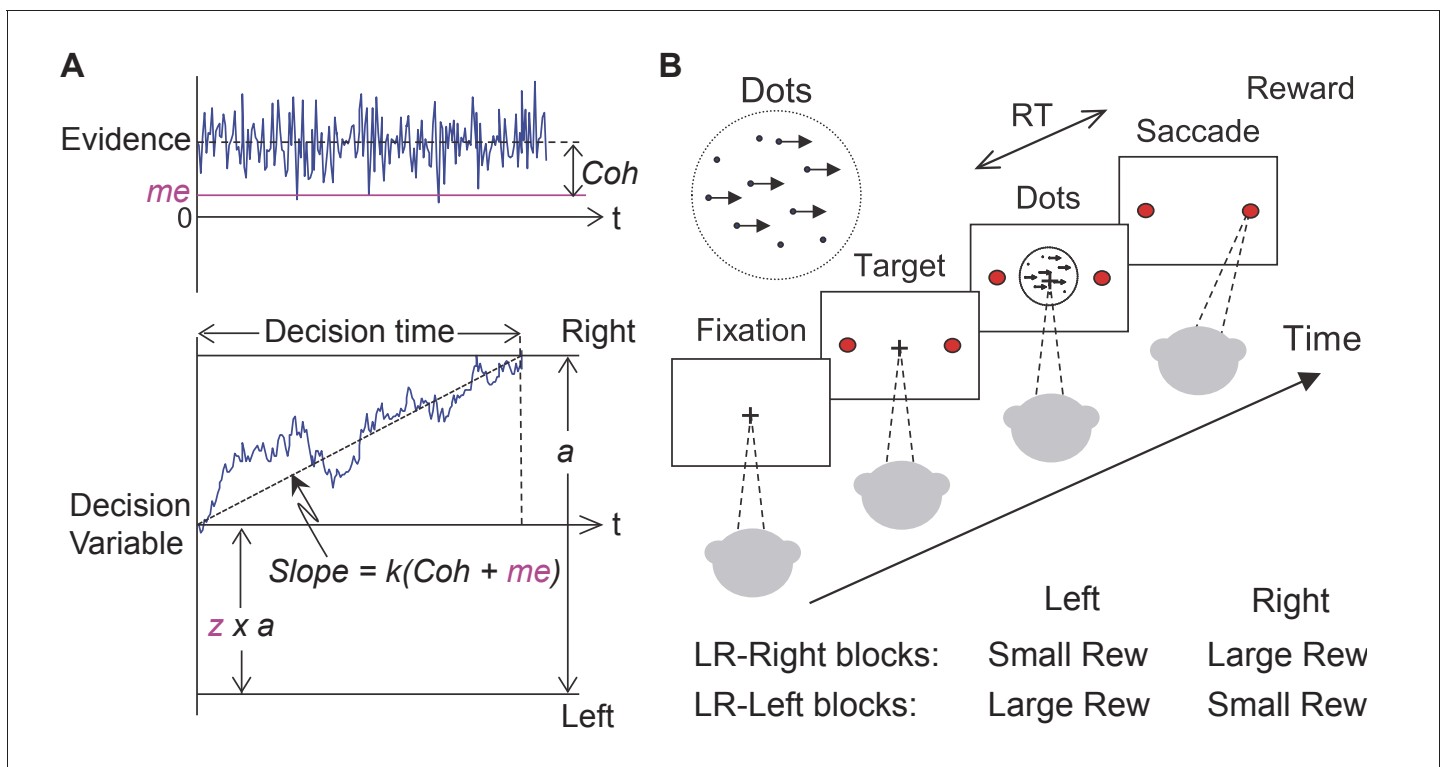

**Figure 1.** Theoretical framework and task design. (**A**) Schematics of the drift-diffusion model (DDM). Motion evidence is modeled as samples from a unit-variance Gaussian distribution (mean: signed coherence, *Coh*). Effective evidence is modeled as the sum of motion evidence and an internal momentary-evidence bias (*me*). The decision variable starts at value a × z, where z governs decision-rule bias and accumulates effective evidence over time with a proportional scaling factor (*k*). A decision is made when the decision variable reaches either bound. Response time (RT) is assumed to be the sum of the decision time and a saccade-specific non-decision time. (**B**) Response-time (RT) random-dot visual motion direction discrimination task with asymmetric rewards. A monkey makes a saccade decision based on the perceived global motion of a random-dot kinematogram. Reward is delivered on correct trials and with a magnitude that depends on reward context. Two reward contexts (LR-Left and LR-Right) were alternated in blocks of trials with signaled block changes. Motion directions and strengths were randomly interleaved within blocks.

# Results

We trained three monkeys to perform the asymmetric-reward random-dot motion discrimination ('dots') task (*Figure 2A*). All three monkeys were initially trained on a symmetric-reward version of the task for which they were required to make fast eye movements (saccades) in the direction congruent with the global motion of a random-dot kinematogram to receive juice reward. They then performed the asymmetric-reward versions that were the focus of this study. Specifically, in blocks of 30 – 50 trials, we alternated direction-reward associations between a 'LR-Right' reward context (the large reward was paired with a correct rightward saccade and the small reward was paired with a correct leftward saccade) and the opposite 'LR-Left' reward context. We also varied the ratio of large versus small reward magnitudes ('reward ratio') across sessions for each monkey. Within a block, we randomly interleaved motion stimuli with different directions and motion strengths (expressed as coherence, the fraction of dots moving in the same direction). We monitored the monkey's choice (which saccade to make) and RT (when to make the saccade) on each trial.

## The monkeys' biases reflected changes in reward context and perceptual sensitivity

For the asymmetric-reward task, all three monkeys tended to make more choices towards the large-reward option, particularly when the sensory evidence was weak. These choice biases corresponded to horizontal shifts in the psychometric function describing the probability of making a rightward choice as a function of signed motion coherence (negative for leftward motion, positive for rightward motion; *Figure 2A*). These functions showed somewhat similar patterns of behavior but some differences in detail for the three monkeys. For example, each monkey showed steady increases in perceptual sensitivity (steepness of the psychometric function), which initially dropped relative to values from the symmetric-reward task then tended to increase with more experience with asymmetric rewards (*Figure 2B*, top; $H_0$: partial Spearman's $\rho$ of sensitivity versus session index after accounting for session-specific reward ratios = 0, p<0.01 in all cases, except LR-Left for monkey C, for which p = 0.56). Moreover, lapse rates were near zero across sessions (*Figure 2B*, bottom), implying that the monkeys knew how to perform the task. The monkeys differed in terms of overall bias, which was the smallest in monkey F. Nevertheless, for all three monkeys bias magnitude tended to decrease over sessions, although this tendency was statistically significant only for monkey C after accounting for co-variations with reward rate (*Figure 2B*, middle). There was often a negative correlation between choice bias and sensitivity, consistent with a general strategy of adjusting bias to obtain more reward (*Figure 2C*; *Figure 2—figure supplement 1C*). Monkeys F and C used suboptimal biases that were larger than the optimal values, whereas monkey A showed greater variations (*Figure 2D*). The monkeys showed only negligible or inconsistent sequential choice biases (*Figure 2—figure supplement 2*), and adding sequential terms did not substantially affect the best-fitting values of the non-sequential terms in the logistic regression (spearman's $\rho$ >0.8 comparing session-by-session best-fitting values of the terms in *Equation 1* with and without additional sequential terms from *Equation 2*). Therefore, all subsequent analyses did not include sequential choice effects.

To better understand the computational principles that governed these idiosyncratic biases, while also taking into account systematic relationships between the choice and RT data, we fit single-trial RT data (i.e., we modeled full RT distributions, not just mean RTs) from individual sessions to a DDM. We used a hierarchical-DDM (HDDM) method that assumes that parameters from individual sessions of the same monkey are samples from a group distribution (*Wiecki et al., 2013*). The HDDM was fit to data from each monkey separately. The HDDM had six parameters for each reward context. Four were from a basic DDM (*Figure 1A*): $a$, the total bound height, representing the distance between the two choice bounds; $k$, a scaling factor that converts sensory evidence (motion strength and direction) to the drift rate; and $t_0$ and $t_1$, non-decision times for leftward and rightward choices, respectively. The additional two parameters provided biases that differed in terms of their effects on the full RT distributions (*Figure 3—figure supplement 1*): $me$, which is additional momentary evidence that is added to the motion evidence at each accumulating step and has asymmetric effects on the two choices and on correct versus error trials (positive values favor the rightward choice); and $z$, which determines the decision rules for the two choices and tends to have asymmetric effects on the two choices but not on correct versus error trials (values > 0.5 favor the rightward

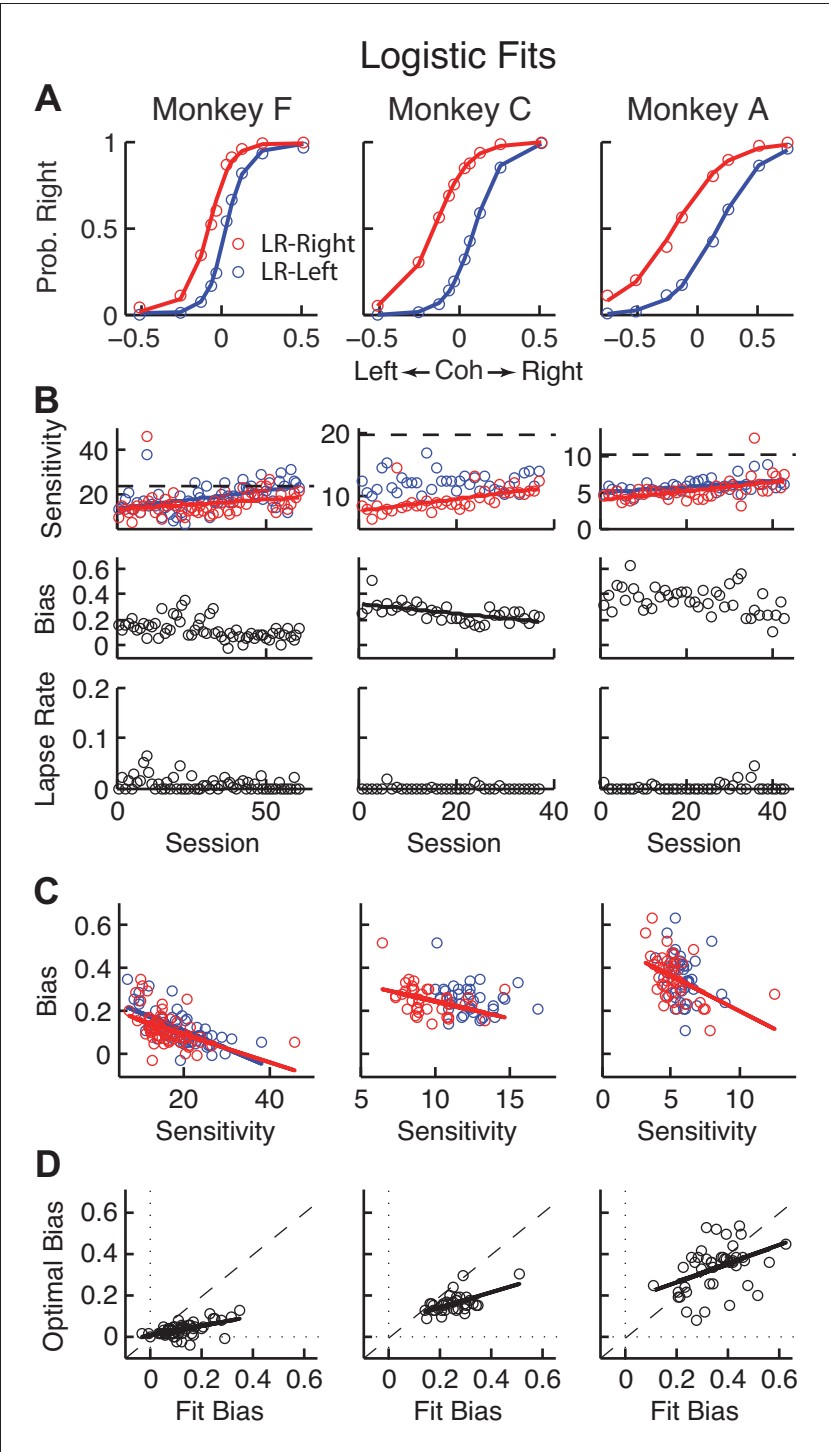

**Figure 2.** Relationships between sensitivity and bias from logistic fits to choice data. (**A**) For each monkey, the probability of making a rightward choice is plotted as a function of signed coherence (–/+indicate left/right motion) from all sessions, separately for the two reward contexts, as indicated. Lines are logistic fits. (**B**) Top row: Motion sensitivity (steepness of the logistic function) in each context as a function of session index (colors as in A). Solid lines indicate significant positive partial Spearman correlation after accounting for changes in reward ratio across sessions ($p<0.05$). Black dashed lines indicate each monkey's motion sensitivity for the task with equal rewards before training on this asymmetric reward task. Middle row: ΔBias (horizontal shift between the two psychometric functions for the two reward contexts at chance level) as a function of session index. Solid line indicates significant negative partial Spearman correlation after accounting for changes in reward ratio across sessions ($p<0.05$). Bottom row: Lapse rate as a function of session index (median = 0 for all three monkeys). (**C**) ΔBias as a function of motion sensitivity for each reward context (colors as in A). Solid line indicates a significant negative partial Spearman correlation after accounting for changes in reward ratio across sessions ($p<0.05$). (**D**) Optimal versus fitted Δbias.

*Figure 2 continued on next page*

*Figure 2 continued*

Optimal Δbias was computed as the difference in the horizontal shift in the psychometric functions in each reward context that would have resulted in the maximum reward per trial, given each monkey's fitted motion sensitivity and experienced values of reward ratio and coherences from each session (see *Figure 2—figure supplement 1*). Solid lines indicate significant positive Spearman correlations (p<0.01). Partial Spearman correlation after accounting for changes in reward ratio across sessions are also significant for moneys F and C (p<0.05).

The online version of this article includes the following source data and figure supplement(s) for figure 2:

**Source data 1.** Task parameters and the monkeys' performance for each trial and each session.
**Source data 2.** Source data for *Figure 2—figure supplement 1*.
**Figure supplement 1.** Relationship between bias and sensitivity.
**Figure supplement 2.** Monkeys showed minimal sequential choice biases.

choice). The HDDM fitting results are shown in *Figure 3*, and summaries of best-fitting parameters and goodness-of-fit metrics are provided in *Table 1*. A DDM variant with collapsing bounds provided qualitatively similar results as the HDDM (*Figure 3—figure supplement 2*). Thus, subsequent analyses use the model with fixed bounds, unless otherwise noted.

The DDM fits provided a parsimonious account of both the choice and RT data. Consistent with the results from the logistic analyses, the HDDM analyses showed that the monkeys made systematic improvements in psychometric sensitivity ($H_0$: partial Spearman's $\rho$ of sensitivity versus session index after accounting for session-specific reward ratios = 0, p<0.01 in all cases except p=0.06 for LR-Left for monkey A). Moreover, there was a negative correlation between psychometric sensitivity and choice bias ($H_0$: partial Spearman's $\rho$ of sensitivity versus total bias after accounting for session-specific reward ratios = 0, p<0.001 in all cases). These fits ascribed the choice biases to changes in both the momentary evidence (*me*) and the decision rule (*z*) of the decision process, as opposed to either parameter alone (*Table 2*). These fits also indicated context-dependent differences in non-decision times, which were smaller for all large-reward choices for all three monkeys except in the LR-Right context for monkeys C and A (*t*-test, p<0.05). However, the differences in non-decision times were relatively small across reward contexts, suggesting that the observed reward biases were driven primarily by effects on decision-related processes.

## The monkeys' bias adjustments were adaptive with respect to optimal reward-rate functions

To try to identify common principles that governed these monkey- and context-dependent decision biases, we analyzed behavior with respect to optimal benchmarks based on certain reward-rate functions. We focused on reward per unit time (RR) and per trial (RTrial), which for this task are optimized in a DDM framework by adjusting momentary-evidence (*me*) and decision-rule (*z*) biases, such that both favor the large-reward choice. However, the magnitudes of these optimal adjustments depend on other task parameters ($a$, $k$, $t_0$, and $t_1$, non-bias parameters from the DDM, plus the ratio of the two reward sizes and inter-trial intervals) that can vary from session to session. Thus, to determine the optimal adjustments, we performed DDM simulations with the fitted HDDM parameters from each session, using different combinations of *me* and *z* values (*Figure 4A*). As reported previously (*Bogacz et al., 2006*; *Simen et al., 2009*), when the large reward was paired with the leftward choice, the optimal strategy used $z < 0.5$ and $me < 0$ (*Figure 4B*, top panels, purple and orange circles for RR and RTrial, respectively). Conversely, when the larger reward was paired with the rightward choice, the optimal strategy used $z > 0.5$ and $me > 0$ (*Figure 4B*, bottom panels).

The monkeys' adjustments of momentary-evidence (*me*) and decision-rule (*z*) biases showed both differences and similarities with respect to these optimal predictions (*Figure 4B*, black circles; similar results were obtained using fits from a model with collapsing bounds, *Figure 4—figure supplement 1*). In the next section, we consider the differences, in particular the apparent use of shifts in *me* in the adaptive direction (i.e., favoring the large-reward choice) but of a magnitude that was larger than predicted, along with shifts in *z* that tended to be in the non-adaptive direction (i.e., favoring the small-reward choice). Here we focus on the similarities and show that the monkeys' decision biases were adaptive with respect to the reward-rate function in four ways (RTrial provided slightly better predictions of the data and thus are presented in the main figures; results based on RR are presented in the Supplementary Figures).

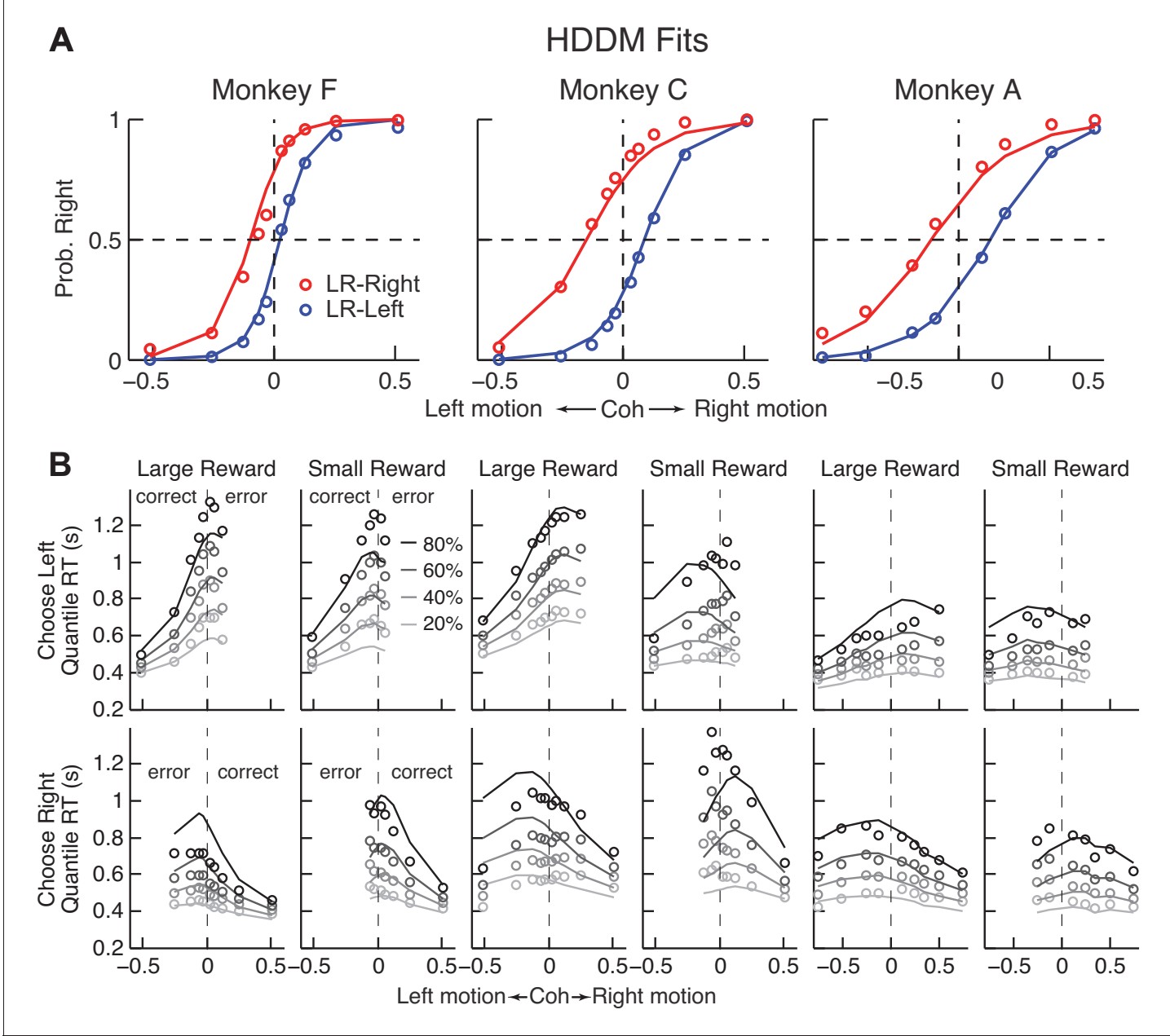

**Figure 3.** Comparison of choice and RT data to HDDM fits with both momentary-evidence (*me*) and decision-rule (*z*) biases. (**A**) Psychometric data (points as in *Figure 2A*) shown with predictions based on HDDM fits to both choice and RT data. B, RT data (circles) and HDDM-predicted RT distributions (lines). Both sets of RT data were plotted as the session-averaged values corresponding to the 20th, 40th, 60th, and 80th percentiles of the full distribution for the five most frequently used coherence levels (we only show data when > 40% of the total sessions contain >4 trials for that combination of motion direction, coherence, and reward context). Top row: Trials in which monkey chose the left target. Bottom row: Trials in which monkeys chose the right target. Columns correspond to each monkey (as in A), divided into choices in the large- (left column) or small- (right column) reward direction (correct/error choices are as indicated in the left-most columns; note that no reward was given on error trials). The HDDM-predicted RT distributions were generated with 50 runs of simulations, each run using the number of trials per condition (motion direction × coherence × reward context × session) matched to experimental data and using the best-fitting HDDM parameters for that monkey.

The online version of this article includes the following source data and figure supplement(s) for figure 3:

**Source data 1.** Source data for *Figure 3—figure supplement 2*.
**Figure supplement 1.** Qualitative comparison between the monkeys' RT distribution and DDM predictions.
**Figure supplement 2.** Fits to a DDM with collapsing bounds.

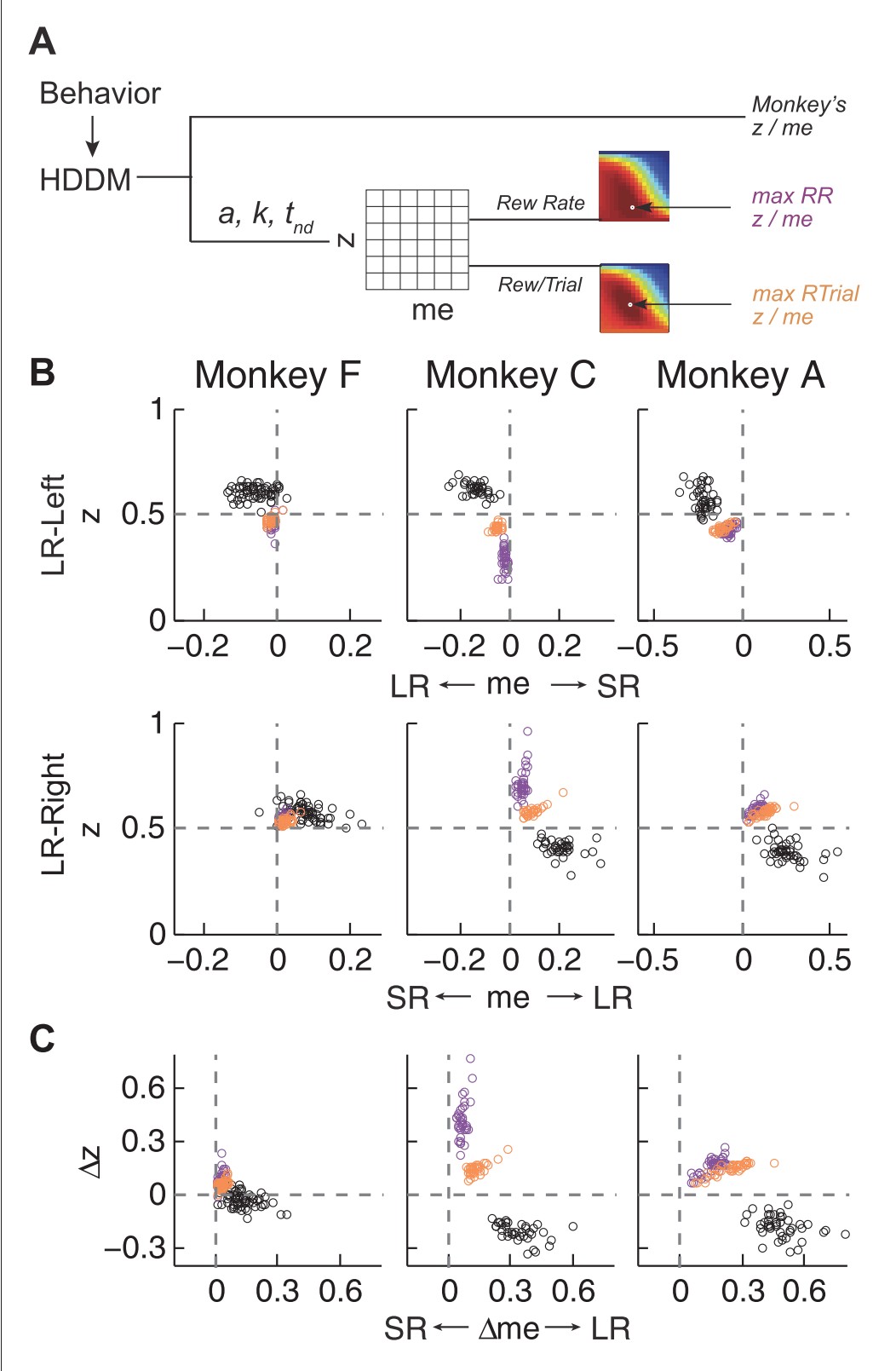

**Figure 4.** Actual versus optimal adjustments of momentary-evidence (*me*) and decision-rule (*z*) biases. (**A**) Schematic of the comparison procedure. Choice and RT data from the two reward contexts in a given session were fitted separately using the HDDM. These context- and session-specific best-fitting *me* and *z* values are plotted as the monkey's data (black circles in B and C). Optimal values were determined by fixing parameters *a*, *k*, and non-

*Figure 4 continued on next page*

*Figure 4 continued*

decision times at best-fitting values from the HDDM and searching in the *me/z* grid space for combinations of *me* and *z* that produced maximal reward function values. For each *me* and *z* combination, the predicted probability of left/right choice and RTs were used with the actual task information (inter-trial interval, error timeout, and reward sizes) to calculate the expected reward rate (RR) and average reward per trial (RTrial). Optimal *me/z* adjustments were then identified to maximize RR (purple) or RTrial (orange). (B) Scatterplots of the monkeys' *me/z* adjustments (black), predicted optimal adjustments for maximal RR (purple), and predicted optimal adjustments for maximal RTrial (orange), for the two reward contexts in all sessions (each data point was from a single session). Values of *me* > 0 or *z* > 0.5 produce biases favoring rightward choices. (C) Scatterplots of the differences in *me* (abscissa) and *z* (ordinate) between the two reward contexts for monkeys (black), for maximizing RR (purple), and for maximizing RTrial (orange). Positive Δ*me* and Δ*z* values produce biases favoring large-reward choices.

The online version of this article includes the following source data and figure supplement(s) for figure 4:

**Source data 1.** RTrial and RR function for each session and reward context.
**Figure supplement 1.** Estimates of momentary-evidence (*me*) and decision-rule (*z*) biases using the collapsing-bound DDM fits.
**Figure supplement 2.** Hypothetical neural activity encoding a reward-biased perceptual decision variable.

First, the best-fitting *me* and *z* values from each monkey corresponded to near-maximal reward rates (**Figure 5A**). We compared the optimal values of reward per trial (RTrial$_{max}$) to the values predicted from the monkeys' best-fitting *me* and *z* adjustments (RTrial$_{predict}$). Both RTrial$_{predict}$ and RTrial$_{max}$ depended on the same non-bias parameters in the HDDM fits that were determined per session ($a$, $k$, $t_0$, and $t_1$) and thus are directly comparable. Their ratios tended to be nearly, but slightly less than, one (mean ratio: 0.977, 0.984, and 0.983 for monkeys F, C, and A, respectively) and remained relatively constant across sessions ($H_0$: slopes of linear regressions of these ratios versus session number = 0, p>0.05 for all three monkeys). Similar results were also obtained using the monkeys' realized rewards, which closely matched RTrial$_{predict}$ (mean ratio: 0.963, 0.980, and 0.974; across-session Spearman's $\rho$ = 0.976, 0.995, and 0.961, for monkeys F, C, and A, respectively, p<0.0001 in all three cases). These results reflected the shallow plateau in the RTrial function near its peak (**Figure 5B**), such that the monkeys' actual adjustments of *me* and *z* were within the contours for 97% RTrial$_{max}$ in most sessions (**Figure 5C**; see **Figure 5—figure supplement 1** for results using RR). Thus, the monkeys' overall choice biases were consistent with strategies that lead to nearly optimal reward outcomes.

Second, the across-session variability of each monkey's decision biases was predicted by idiosyncratic features of the reward functions. The reward functions were, on average, different for the two reward contexts and each of the three monkeys (**Figure 6A**). These differences included the size of the near-maximal plateau (red patch), which determined the level of tolerance in RTrial for deviations from optimal adjustments in *me* and *z*. This tolerance corresponded to the session-by-session variability in each monkey's *me* and *z* adjustments (**Figure 6B**). In general, monkey F had the smallest plateaus and tended to use the narrowest range of *me* and *z* adjustments across sessions. In contrast, monkey A had the largest plateaus and tended to use the widest range of *me* and *z* adjustments (Pearson's $\rho$ between the size of the 97% RTrial contour, in pixels, and the sum of the across-session variances in each monkeys' *me* and *z* adjustments = 0.83, p=0.041). Analyses using the RR function produced qualitatively similar results (**Figure 6—figure supplement 1**).

Third, the session-by-session adjustments in both *me* and *z* corresponded to particular features of each monkey's context-specific reward function. The shape of this function, including the orientation of the plateau with respect to *z* and *me*, depended on the monkey's perceptual sensitivity and the reward ratio for the given session. The monkeys' *me* and *z* adjustments varied systematically with this orientation (**Figure 6C and D** for RTrial, **Figure 6—figure supplement 1C and D** for RR). This result was not an artifact of the fitting procedure, which was able to recover appropriate, simulated bias parameter values regardless of the values of non-bias parameters that determine the shape of the reward function (**Figure 6—figure supplement 2**).

Fourth, the monkeys' *me* and *z* adjustments were correlated with the values that would maximize RTrial, given the value of the other parameter for the given session and reward context (**Figure 6E** for RTrial, **Figure 6—figure supplement 1E** for RR). These correlations were substantially weakened by shuffling the session-by-session reward functions (**Figure 6—figure supplement 3**). Together, these results suggest that all three monkeys used biases that were adaptively calibrated with respect to the reward information and perceptual sensitivity of each session.

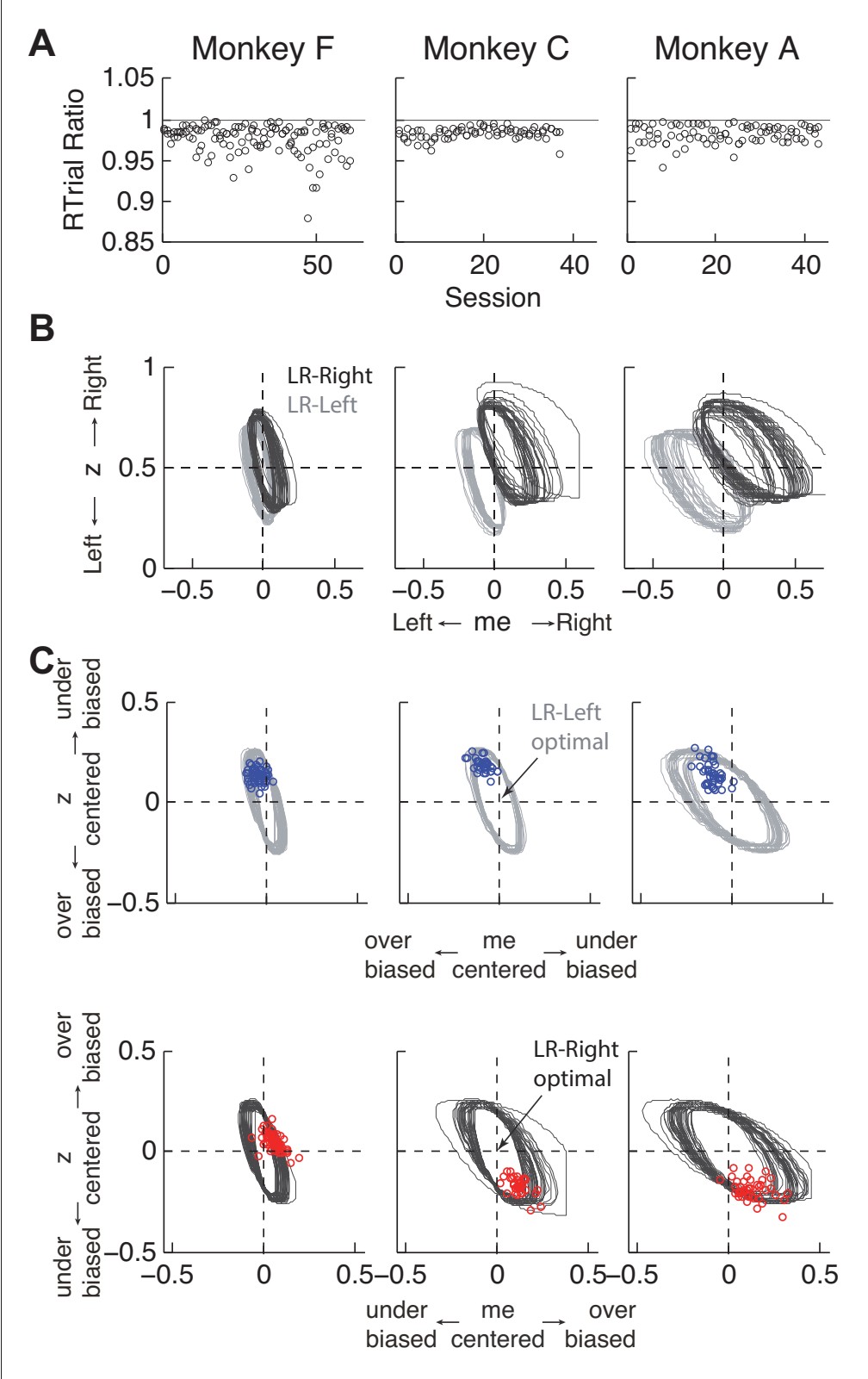

**Figure 5.** Predicted versus optimal reward per trial (RTrial). (**A**) Scatterplots of RTrial$_{predict}$:RTrial$_{max}$ ratio as a function of session index. Each session was represented by two ratios, one for each reward context. (**B**) 97% RTrial$_{max}$ contours for all sessions, computed using the best-fitting HDDM parameters and experienced coherences and reward ratios from each session. Light grey: LR-Left blocks; Dark grey: LR-Right blocks. (**C**) The monkeys' adjustments

*Figure 5 continued on next page*

*Figure 5 continued*

(blue: LR-Left blocks, red: LR-Right blocks) were largely within the 97% RTrial$_{max}$ contours for all sessions and tended to cluster in the *me* over-biased, *z* under-biased quadrants (except Monkey F in the LR-Right blocks). The contours and monkeys' adjustments are centered at the optimal adjustments for each session.

The online version of this article includes the following figure supplement(s) for figure 5:

**Figure supplement 1.** Predicted versus optimal reward rate (RR).

## The monkeys' adaptive adjustments were consistent with a satisficing, gradient-based learning process

Thus far, we showed that all three monkeys adjusted their decision strategies in a manner that matched many features of the optimal predictions based on their idiosyncratic, context-specific reward-rate functions. However, their biases did not match the optimal predictions exactly. Specifically, all three monkeys used shifts in *me* favoring the large-reward choice (adaptive direction) but of a magnitude that was larger than predicted, along with shifts in *z* favoring the small-reward choice (non-adaptive direction). We next show that these shifts can be explained by a model in which the monkeys are initially over-biased, then adjust their model parameters to increase reward and stop learning when the reward is high enough, but not at its maximum possible value.

The intuition for this gradient-based satisficing model is shown in *Figure 7*. The lines on the RTrial heatmap represent the trajectories of a gradient-tracking procedure that adjusts *me* and *z* values to increase RTrial until a termination point (for illustration, here we used 97% of the maximum possible value). For example, consider adjusting *me* and *z* by following all of the magenta gradient lines until their end-points. The lines are color-coded by *me/z* being adaptive vs. non-adaptive, regardless of their relative magnitudes to the optimal values. In other words, as long as the initial *me* and *z* values fall within the area covered by the magenta lines, the positive gradient-tracking procedure would lead to a good-enough solution with adaptive *me* and non-adaptive *z* values similar to what we found in the monkeys' data. *Figure 7* also illustrates why assumptions about the starting point of this adaptive process are important: randomly selected starting points would result in learned *me* and *z* values distributed around the peak of the reward function, whereas the data (e.g., *Figure 5C*) show distinct clustering that implies particular patterns of starting points.

We simulated this process using: (1) different starting points; (2) gradients defined by the reward function derived separately for each reward context, session, and monkey; and (3) a termination rule corresponding to achieving each monkey's average reward in that session (RTrial$_{predict}$) estimated from the corresponding best-fitting model parameters and task conditions. This process is illustrated for LR-Left blocks in an example session from monkey C (*Figure 8A*). We estimated the unbiased *me* and *z* values as the midpoints between their values for LR-Left and LR-Right blocks (square). At this point, the RTrial gradient is larger along the *me* dimension than the *z* dimension, reflecting the tilt of the reward function. We set the initial point at baseline *z* and a very negative value of *me* (90% of the highest coherence used in the session; overshoot in the adaptive direction) and referred to this setting as the 'over-*me*' model. The *me* and *z* values were then updated according to the RTrial gradient (see cartoon insert in *Figure 8A*), until the monkey's RTrial$_{predict}$ or better was achieved (magenta trace and circle). The endpoint of this updating process was very close to monkey C's actual adjustment (gray circle). For comparison, three alternative models are illustrated. The 'over-*z*' model assumes updating from the baseline *me* and over-adjusted *z* values (blue, initial *z* set as 0.1 for the LR-Left context and 0.9 for the LR-Right context). The 'over-both' model assumes updating from the over-adjusted *me* and *z* values (green). The 'neutral' model assumes the same updating process but from the baseline *me* and baseline *z* (black). The endpoints from these alternative models deviated considerably from the monkey's actual adjustment.

The 'over-*me*' model produced better predictions than the other three alternative models for all three monkeys. Of the four models, only the 'over-*me*' model captured the monkeys' tendency to bias *me* toward the large-reward choice (positive Δ*me*) and bias *z* toward the small-reward choice (negative Δ*z*; *Figure 8B*). In contrast, the 'over-*z*' model predicted small adjustments in *me* and large adjustments in *z* favoring the large-reward choice; the 'over-both' model predicted relatively large, symmetric *me* and *z* adjustments favoring the large-reward choice; and the 'neutral' model predicted relatively small, symmetric adjustments in both *me* and *z* favoring the large-reward choice. Accordingly, for each monkey, the predicted and actual values of both Δ*me* and Δ*z* were most strongly

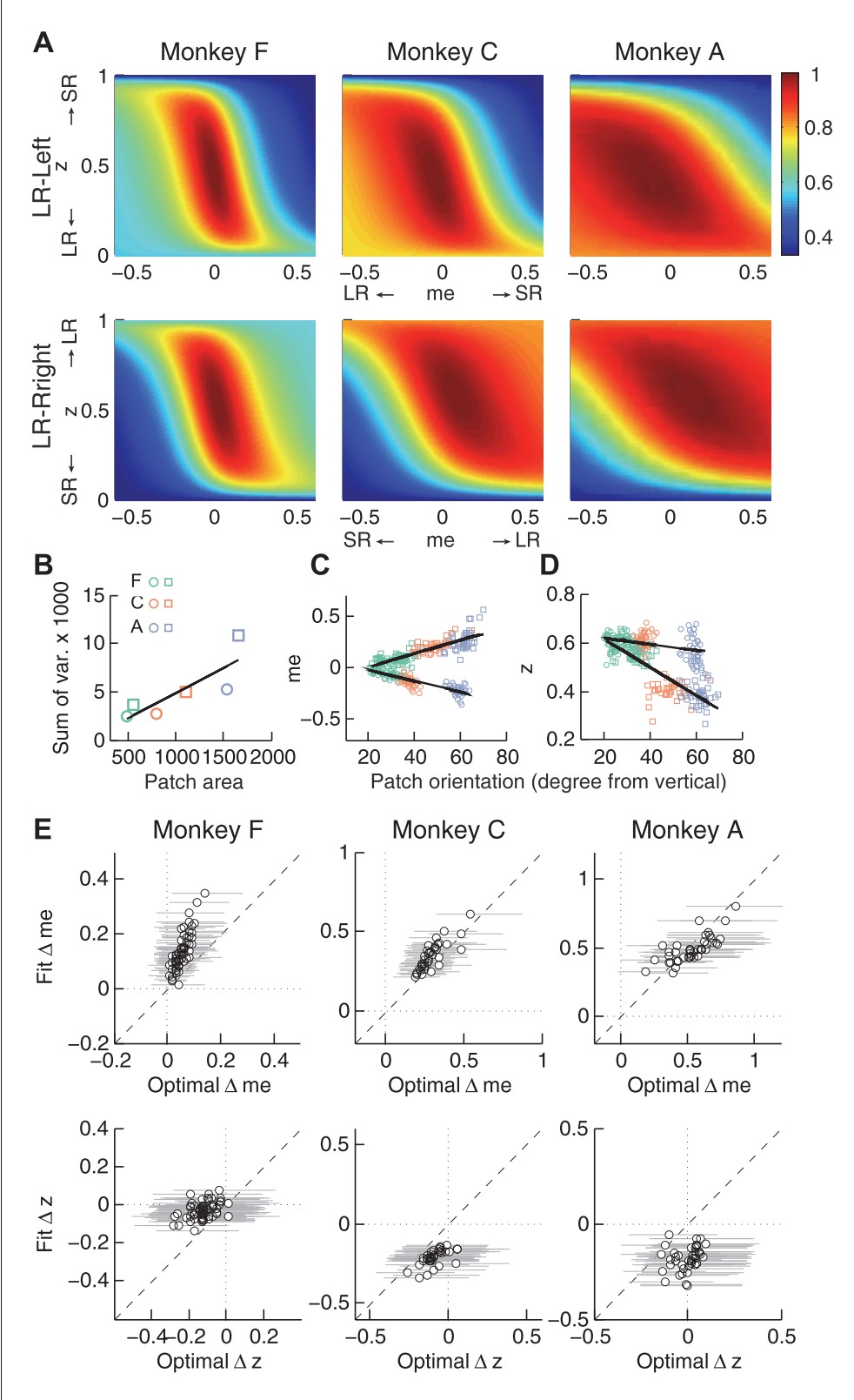

**Figure 6.** Relationships between adjustments of momentary-evidence (*me*) and decision-rule (*z*) biases and RTrial function properties. (**A**) Mean RTrial as a function of *me* and *z* adjustments for the LR-Left (top) and LR-Right (bottom) blocks. Hotter colors represent larger RTrial values (see legend to the right). RTrial was normalized to RTrial$_{max}$ for each session and then averaged across sessions. (**B**) Scatterplot of the total variance in *me* and *z*

*Figure 6 continued on next page*

*Figure 6 continued*

adjustments across sessions (ordinate) and the area of >97% max of the average RTrial patch (abscissa). Variance and patch areas were measured separately for the two reward blocks (circles for LR-Left blocks, squares for LR-Right blocks). (C, D) The monkeys' session- and context-specific values of *me* (C) and *z* (D) co-varied with the orientation of the >97% heatmap patch (same as the contours in **Figure 5B**). Orientation is measured as the angle of the tilt from vertical. Circles: data from LR-Left block; squares: data from LR-Right block; lines: significant correlation between *me* (or *z*) and patch orientations across monkeys (p<0.05). Colors indicate different monkeys (see legend in B). E, Scatterplots of conditionally optimal versus fitted Δ*me* (top row) and Δ*z* (bottom row). For each reward context, the conditionally optimal *me* (*z*) value was identified given the monkey's best-fitting *z* (*me*) values. The conditionally optimal Δ*me* (Δ*z*) was the difference between the two conditional optimal *me* (*z*) values for the two reward contexts. Grey lines indicate the range of conditional Δ*me* (Δ*z*) values corresponding to the 97% maximal RTrial given the monkeys' fitted *z* (*me*) values.

The online version of this article includes the following figure supplement(s) for figure 6:

**Figure supplement 1.** The monkeys' momentary-evidence (*me*) and decision-rule (*z*) adjustments reflected RR function properties.
**Figure supplement 2.** The HDDM model fitting procedure does not introduce spurious correlations between patch orientation and *me* value.
**Figure supplement 3.** The correlation between fitted and conditionally optimal adjustments was stronger for the real, session-by-session data (red lines) than for unmatched (shuffled) sessions (bars).

positively correlated for predictions from the 'over-*me*' model compared to the other models (**Figure 8C**). The 'over-*me*' model was also the only one of the models we tested that recapitulated the measured relationships between both *me*- and *z*-dependent biases and session-by-session changes in the orientation of the RTrial function (**Figure 8D**). Similar results were observed using RR function (**Figure 7—figure supplement 1** and **Figure 8—figure supplement 1**). We also examined whether the shape of the reward surface alone can explain the monkeys' bias patterns. We repeated the simulations using randomized starting points, with or without additional noise in each updating step. These simulations could not reproduce the monkeys' bias patterns (data not shown), suggesting that using 'over-*me*' starting points is critical for accounting for the monkeys' suboptimal behavior.

## Discussion

We analyzed the behavior of three monkeys performing a decision task that encouraged the use of both uncertain visual motion evidence and the reward context. All three monkeys made choices that were sensitive to the strength of the sensory evidence and were biased toward the larger-reward choice, which is roughly consistent with results from previous studies of humans and monkeys performing similar tasks (**Maddox and Bohil, 1998**; **Voss et al., 2004**; **Diederich and Busemeyer, 2006**; **Liston and Stone, 2008**; **Serences, 2008**; **Feng et al., 2009**; **Simen et al., 2009**; **Nomoto et al., 2010**; **Summerfield and Koechlin, 2010**; **Teichert and Ferrera, 2010**; **Gao et al., 2011**; **Leite and Ratcliff, 2011**; **Mulder et al., 2012**; **Wang et al., 2013**; **White and Poldrack, 2014**). However, we also found that these adjustments differed considerably in detail for the three monkeys, in terms of overall magnitude, dependence on perceptual sensitivity and offered rewards, and relationship to RTs. We quantified these effects with a logistic analysis and a commonly used model of decision-making, the drift-diffusion model (DDM), which allowed us to compare the underlying decision-related computations to hypothetical benchmarks that would maximize reward. We found that all three monkeys made reward context-dependent adjustments with two basic components: (1) an over-adjustment of the momentary evidence provided by the sensory stimulus (*me*) in favor of the large-reward option; and (2) an adjustment to the decision rule that governs the total evidence needed for each choice (*z*), but in the opposite direction (i.e., towards the small-reward option). Similar to some earlier reports of human and monkey performance on somewhat similar tasks, our monkeys did not optimize reward rate (**Starns and Ratcliff, 2010Starns and Ratcliff, 2012**; **Teichert and Ferrera, 2010**). Instead, their adjustments tended to provide nearly, but not exactly, maximal reward intake. We proposed a common heuristic strategy based on the monkeys' individual reward functions to account for the idiosyncratic adjustments across monkeys and across sessions within the same monkey.

### Considerations for assessing optimality and rationality

Assessing decision optimality requires a model of the underlying computations. In this study, we chose the DDM for several reasons. First, it provided a parsimonious account of both the choice and RT data (**Palmer et al., 2005**; **Ratcliff et al., 1999**). Second, as discussed in more detail below, the

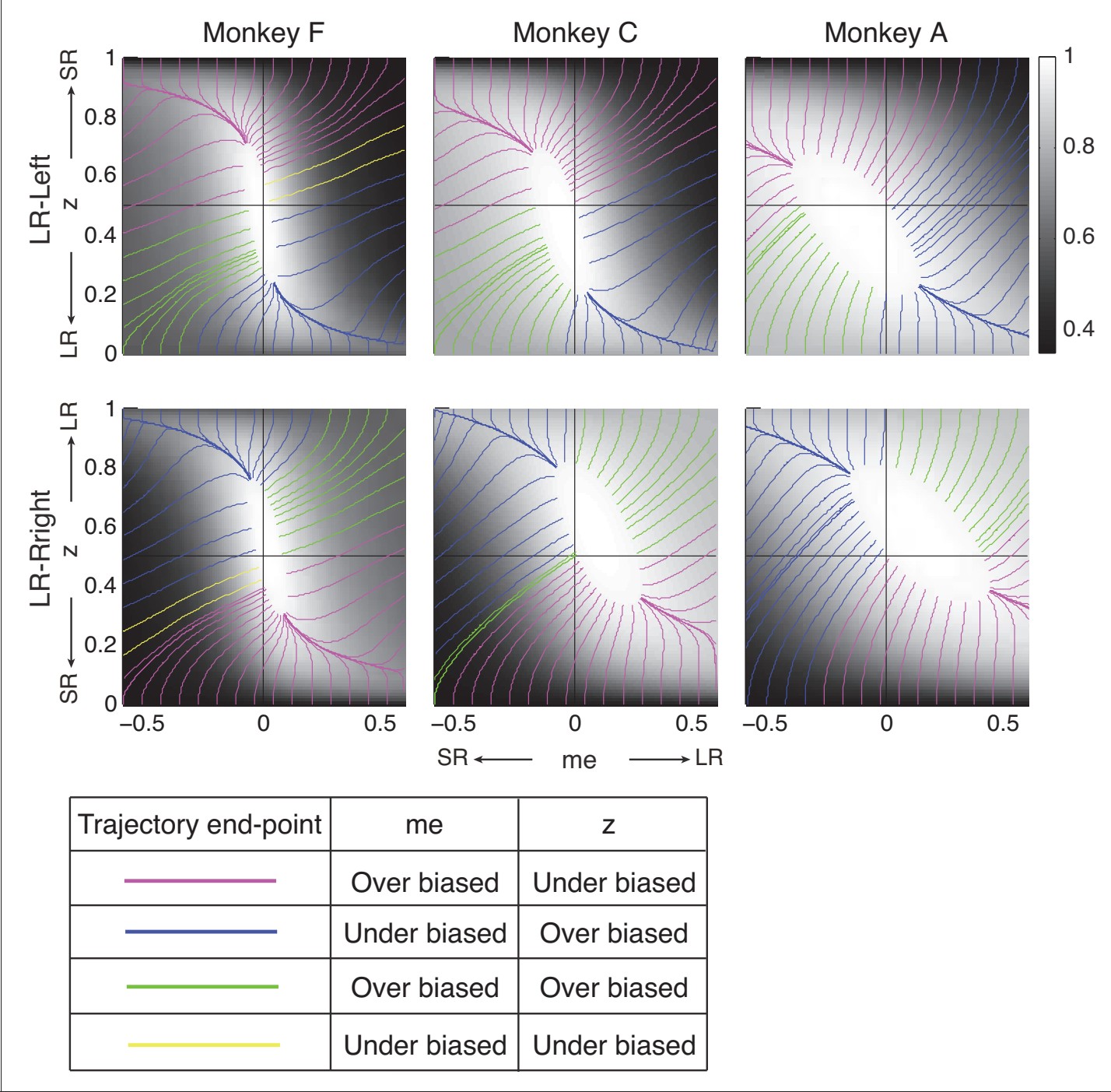

**Figure 7.** Relationships between starting and ending values of the satisficing, reward function gradient-based updating process. Example gradient lines of the average RTrial maps for the three monkeys are color coded based on the end point of gradient-based *me* and *z* adjustments in the following ways: (1) *me* biases to large reward whereas *z* biases to small reward (magenta); (2) *z* biases to large reward whereas *me* biases to small reward (blue); (3) *me* and *z* both bias to large reward (green), and (4) *me* and *z* both bias to small reward (yellow). The gradient lines ended on the 97% $RTrial_{max}$ contours. Top row: LR-Left block; bottom row: LR-Right block.

The online version of this article includes the following figure supplement(s) for figure 7:

**Figure supplement 1.** RR gradient trajectories color-coded by the end points of the *me/z* patterns.

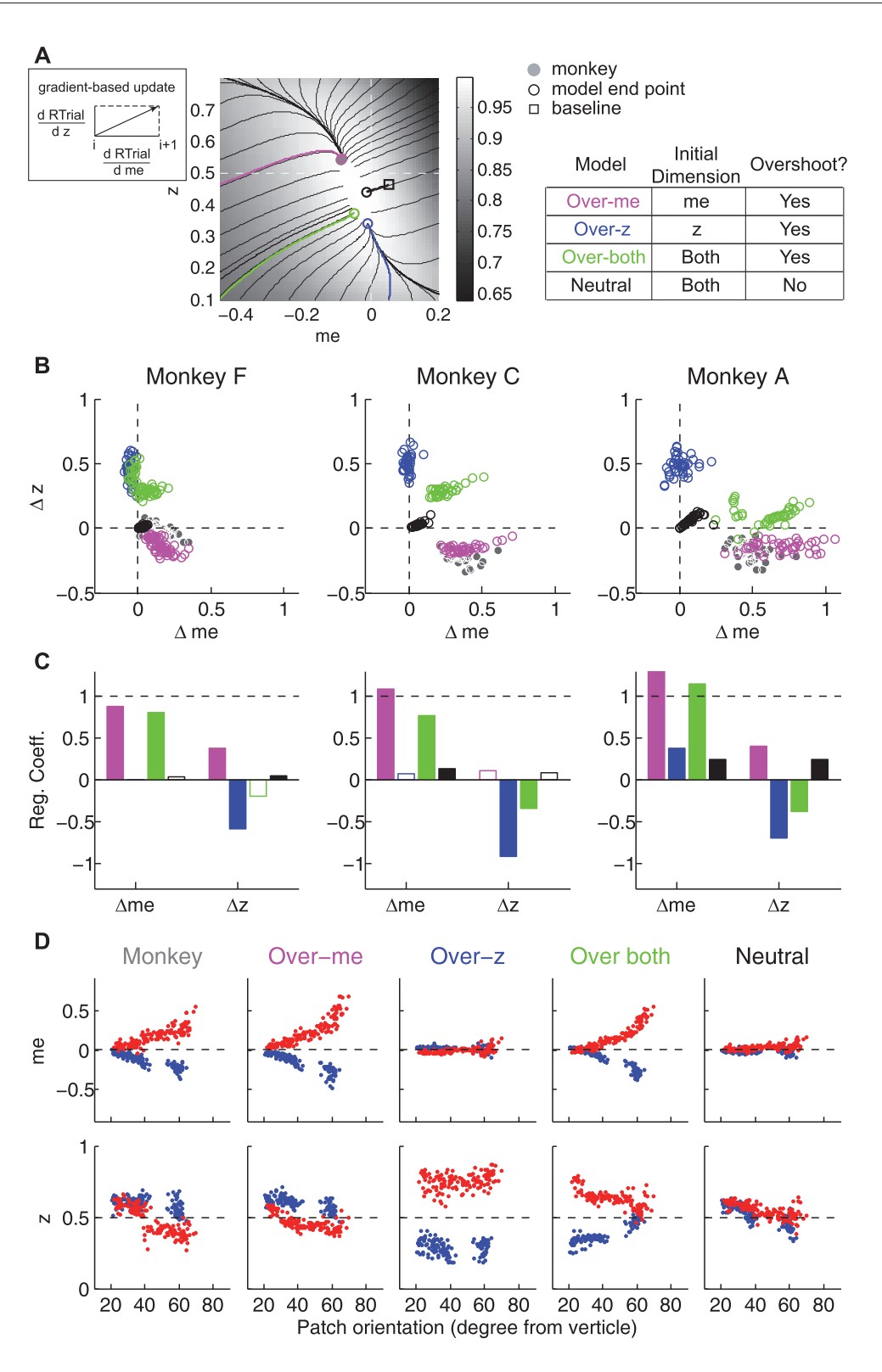

**Figure 8.** The satisficing reward function gradient-based model. (**A**) Illustration of the procedure for predicting a monkey's *me* and *z* values for a given RTrial function. For better visibility, RTrial for the LR-Left reward context in an example session is shown as a heatmap in greyscale. Gradient lines are shown as black lines. The square indicates the unbiased *me* and *z* combination (average values across the two reward contexts). The four trajectories

*Figure 8 continued on next page*

*Figure 8 continued*

represent gradient-based searches based on four alternative assumptions of initial values (see table on the right). All four searches stopped when the reward exceeded the average reward the monkey received in that session (RTrial$_{predict}$), estimated from the corresponding best-fitting model parameters and task conditions. Open circles indicate the end values. Grey filled circle indicates the monkey's actual *me* and z. Note that the end points differ among the four assumptions, with the magenta circle being the closest to the monkey's fitted *me* and z of that session. (**B**) Scatterplots of the predicted and actual Δ*me* and Δz between reward contexts. Grey circles here are the same as the black circles in *Figure 4C*. Colors indicate model identity, as in (**A**). (**C**) Average regression coefficients between each monkey's Δ*me* (left four bars) and Δz (right four bars) values and predicted values for each of the four models. Filled bars: *t*-test, p<0.05. (**D**) Covariation of *me* (top) and z (bottom) with the orientation of the >97% maximal RTrial heatmap patch for monkeys and predictions of the four models. Blue: data from LR-Left blocks, red: data from LR-Right blocks. Data in the 'Monkey' column are the same as in *Figure 6C and D*. Note that predictions of the 'over-*me*' model best matched the monkey data than the other models.

The online version of this article includes the following figure supplement(s) for figure 8:

**Figure supplement 1.** Predictions of a RR gradient-based model.
**Figure supplement 2.** Dependence of the orientation and area of the near-optimal RTrial patch on parameters reflecting internal decision process and external task specifications.
**Figure supplement 3.** The joint effect of DDM model parameters *a* (governing the speed-accuracy trade-off) and *k* (governing perceptual sensitivity) on the shape of the reward function.
**Figure supplement 4.** Effects of the shape of the reward function on deviations from optimality.

DDM and related accumulate-to-bound models have provided useful guidance for identifying neural substrates of the decision process (*Roitman and Shadlen, 2002*; *Ding and Gold, 2010*; *Ding and Gold, 2012a*; *Hanks et al., 2011*; *Ratcliff et al., 2003*; *Rorie et al., 2010*; *Mulder et al., 2012*; *Summerfield and Koechlin, 2010*; *Frank et al., 2015*). Third, these models are closely linked to normative theory, including under certain assumptions matching the statistical procedure known as the sequential probability ratio test that can optimally balance the speed and accuracy of uncertain decisions (*Barnard, 1946*; *Wald, 1947*; *Wald and Wolfowitz, 1948*; *Edwards, 1965*). These normative links were central to our ability to use the DDM to relate the monkeys' behavior to different forms of reward optimization. The particular form of DDM that we used produced reasonably good, but not perfect, fits to the monkeys' data. These results support the utility of the DDM framework but also underscore the fact that we do not yet know the true model, which could impact our optimality assessment.

Assessing optimality also requires an appropriate definition of the optimization goal. In our study, we focused primarily on the goal of maximizing reward rate (per trial or per unit of time). Based on this definition, the monkeys showed suboptimal reward-context-dependent adjustments. It is possible that the monkeys' were optimizing for a different goal, such as accuracy or a competition between reward and accuracy ('COBRA,' *Maddox and Bohil, 1998*). However, the monkeys' behavior was not consistent with optimizing for these goals, either. Specifically, none of these goals would predict optimal z adjustment that favors the small reward choice: accuracy maximization would require unbiased decisions (*me* = 0 and z = 0.5), whereas COBRA would require z values with smaller magnitude (between 0.5 and those predicted for reward maximization alone), but still in the adaptive direction. Therefore, the monkeys' strategies were not consistent with simply maximizing commonly considered reward functions.

Deviations from optimal behavior are often ascribed to a lack of effort or poor learning. However, these explanations seem unlikely to be primary sources of suboptimality in our study. For example, lapse rates, representing the overall ability to attend to and perform the task, were consistently near zero for all three monkeys. Moreover, the monkeys' reward outcomes (RTrial or RR with respect to optimal values) did not change systematically with experience but instead stayed close to the optimal values. These results imply that the monkeys understood the task demands and performed consistently well over the course of our study. Suboptimal performance has also been observed in human subjects, even with explicit instructions about the optimality criteria (*Starns and Ratcliff, 2010*; *Starns and Ratcliff, 2012*), suggesting that additional factors need to be considered to understand apparent suboptimality in general forms of decision-making. In our study, the monkeys made adjustments that were adapted to changes in their idiosyncratic, context-dependent reward functions, which reflected session-specific reward ratios and motion coherences and the monkeys' daily variations of perceptual sensitivity and speed-accuracy trade-offs (*Figure 6*, *Figure 6—figure supplement 1*). Based on these observations, we reasoned that the seemingly sub-optimal behaviors

may instead reflect a common, adaptive, rational strategy that aimed to attain good-enough (satisficing) outcomes.

The gradient-based, satisficing model we proposed was based on the considerations discussed below to account for our results. We do not yet know how well this model generalizes to other tasks and conditions, but it exemplifies an additional set of general principles for assessing the rationality of decision-making behavior: goals that are not necessarily optimal but good enough, potential heuristic strategies based on the properties of the utility function, and flexible adaptation to changes in the external and internal conditions.

## Assumptions and experimental predictions of the proposed learning strategy

In general, finding rational solutions through trial-and-error or stepwise updates requires a sufficient gradient in the utility function to drive learning (*Sutton and Barto, 1998*). Our proposed scheme couples a standard gradient-following algorithm with principles that have been used to explain and facilitate decisions with high uncertainties, time pressures, and/or complexity to achieve a satisficing solution (*Simon, 1966*; *Wierzbicki, 1982*; *Gigerenzer and Goldstein, 1996*; *Nosofsky and Palmeri, 1997*; *Goodrich et al., 1998*; *Sakawa and Yauchi, 2001*; *Goldstein and Gigerenzer, 2002*; *Stirling, 2003*; *Gigerenzer, 2010*; *Oh et al., 2016*). This scheme complements but differs from a previously proposed satisficing strategy to account for human subjects' suboptimal calibration of the speed-accuracy trade-off via adjustments of the decision bounds of a DDM that favor robust solutions given uncertainties about the inter-trial interval (*Zacksenhouse et al., 2010*). In contrast, our proposed strategy focuses on reward-biased behaviors for a given speed-accuracy tradeoff and operates on reward per trial, which is, by definition, independent of inter-trial-interval.

Our scheme was based on four key assumptions, as follows. Our first key assumption was that the starting point for gradient following was not the unbiased state (i.e., $me = 0$ and $z = 0.5$) but an over-biased state. Notably, in many cases the monkeys could have performed as well or better than they did, in terms of optimizing reward rate, by making unbiased decisions. The fact that none did so prompted our assumption that their session-by-session adjustments tended to reduce, not inflate, biases. Specifically, we assumed that the initial experience of the asymmetric reward prompted an over-reaction to bias choices towards the large-reward alternative. In general, such an initial over-reaction is not uncommon, as other studies have shown excessive, initial biases that are reduced or eliminated with training (*Gold et al., 2008*; *Jones et al., 2015*; *Nikolaev et al., 2016*). The over-reaction is also rational because the penalty is larger for an under-reaction than for an over-reaction. For example, in the average RTrial heatmaps for our task (*Figure 6A*), the gradient dropped faster in the under-biased side than in the over-biased side. This pattern is generally true for tasks with sigmoid-like psychometric functions (for example, the curves in *Figure 2—figure supplement 1*). Our model further suggests that the nature of this initial reaction, which may be driven by individually tuned features of the reward function that can remain largely consistent even for equal-reward tasks (*Figure 8—figure supplement 2*) and then constrain the end-points of a gradient-based adjustment process (*Figure 8*), may help account for the extensive individual variability in biases that has been reported for reward-biased perceptual tasks (*Voss et al., 2004*; *Summerfield and Koechlin, 2010*; *Leite and Ratcliff, 2011*; *Cicmil et al., 2015*).

The specific form of initial over-reaction in our model, which was based on the gradient asymmetry of the reward function, makes testable predictions. Specifically, our data were most consistent with an initial bias in momentary evidence (*me*), which caused the biggest change in the reward function. However, this gradient asymmetry can change dramatically under different conditions. For example, changes in the subject's cautiousness (i.e., the total bound height parameter, *a*) and perceptual sensitivity (*k*) would result in a steeper gradient in the other dimension (the decision rule, or *z*) of the reward function (*Figure 8—figure supplement 3*). Our model predicts that such a subject would be more prone to an initial bias along that dimension. This prediction can be tested by using speed-accuracy instructions to affect the bound height and different stimulus parameters to change perceptual sensitivity (*Palmer et al., 2005*; *Gegenfurtner and Hawken, 1996*).

Our second key assumption was that from this initial, over-biased state, the monkeys made adjustments to both the momentary evidence (*me*) and decision rule (*z*) that generally followed the gradient of the reward function. The proposed step-wise adjustments occurred too quickly to be evident in behavior; for example the estimated biases were similar for the early and late halves in a

block (data not shown). Instead, our primary support for this scheme was that the steady-state biases measured in each session were tightly coupled to the shape of the reward function for that session. It would be interesting to design tasks that might allow for more direct measurements of the updating process itself, for example, by manipulating both the initial biases and relevant reward gradient that might promote a longer adjustment process.

Our third key assumption was that the shallowness of the utility of the function around the peak supported satisficing solutions. Specifically, gradient-based adjustments, particularly those that use rapid updates based on implicit knowledge of the utility function, may be sensitive only to relatively large gradients. For our task, the gradients were much smaller around the peak, implying that there were large ranges of parameter values that provided such similar outcomes that further adjustments were not used. In principle, it is possible to change the task conditions to test if and how subjects might optimize with respect to steeper functions around the peak. For example, for RTrial, the most effective way to increase the gradient magnitude near the peak (i.e., reducing the area of the dark red patch) is to increase sensory sensitivity ($k$) or cautiousness ($a$; i.e., emphasizing accuracy over speed; *Figure 8—figure supplement 2*). For RR, the gradient can also be enhanced by increasing the time-out penalty. Despite some practical concerns about these manipulations (e.g., increasing time-out penalties can decrease motivation), it would be interesting to study their effects on performance in more detail to understand the conditions under which satisficing or 'good enough' strategies are used (*Simon, 1956*; *Simon, 1982*).

Our fourth key assumption was that the monkeys terminated adjustments as soon as they reached a good-enough reward outcome. This termination rule produced end points that approximated the monkeys' behavior reasonably well. Other termination rules are likely to produce similar end points. For example, the learning rate for synaptic weights might decrease as the presynaptic and postsynaptic activities become less variable (*Aitchison et al., 2017*; *Kirkpatrick et al., 2017*). In this scheme, learning gradually slows down as the monkey approaches the plateau on the reward surface, which might account for our results.

The satisficing reward gradient-based scheme we propose may further inform appropriate task designs for future studies. For example, our scheme implies that the shape of the reward function near the peak, particularly the steepness of the gradient, can have a strong impact on how closely a subject comes to the optimal solution for a given set of conditions. Thus, task manipulations that affect the shape of the reward-function peak could, in principle, be used to control whether a study focuses on more- or less-optimal behaviors (*Figure 8—figure supplement 4*). For example, increasing perceptual sensitivity (e.g., via training) and/or decisions that emphasize accuracy over speed (e. g., via instructions) tends to sharpen the peak of the reward function. According to our scheme, this sharpening should promote increasingly optimal decision-making, above and beyond the performance gains associated with increasing accuracy, because the gradient can be followed closer to the peak of the reward function. The shape of the peak is also affected by the reward ratio, such that higher ratios lead to larger plateaus, i.e. shallower gradient, near the peak. This relationship leads to the idea that, all else being equal, a smaller reward ratio may be more suitable for investigating principles of near-optimal behavior, whereas a larger reward ratio may be more suitable for investigating the source and principles of sub-optimal behaviors.

## Possible neural mechanisms

The DDM framework has been used effectively to identify and interpret neural substrates of key computational components of the decision process for symmetric-reward versions of the motion-discrimination task. Our study benefitted from an RT task design that provided a richer set of constraints for inferring characteristics of the underlying decision process than choice data alone (*Feng et al., 2009*; *Nomoto et al., 2010*; *Teichert and Ferrera, 2010*). The monkeys' strategy further provides valuable anchors for future studies of the neural mechanisms underlying decisions that are biased by reward asymmetry, stimulus probability asymmetry, and other task contexts.

For neural correlates of bias terms in the DDM, it is commonly hypothesized that $me$ adjustments may be implemented as modulation of MT output and/or synaptic weights for the connections between different MT subpopulations and decision areas (*Cicmil et al., 2015*). In contrast, $z$ adjustments may be implemented as context-dependent baseline changes in neural representations of the decision variable and/or context-dependent changes in the rule that determines the final choice (*Lo and Wang, 2006*; *Rao, 2010*; *Lo et al., 2015*; *Wei et al., 2015*). The manifestation of these

adjustments in neural activity that encodes a decision variable may thus differ in its temporal characteristics: a $me$ adjustment is assumed to modulate the rate of change in neural activity, whereas a $z$ adjustment does not. However, such a theoretical difference can be challenging to observe, because of the stochasticity in spike generation and, given such stochasticity, practical difficulties in obtaining sufficient data with long decision deliberation times. By adjusting $me$ and $z$ in opposite directions, our monkeys' strategies may allow a simpler test to disambiguate neural correlates of $me$ and $z$. Specifically, a neuron or neuronal population that encodes $me$ may show reward modulation congruent with its choice preference, whereas a neuron or neuronal population that encodes $z$ may show reward modulation opposite to its choice preference (*Figure 4—figure supplement 2*). These predictions further suggest that, although it is important to understand if and how human or animal subjects can perform a certain task optimally, for certain systems-level questions, there may be benefits to tailoring task designs to promote sub-optimal strategies in otherwise well-trained subjects.

## Materials and methods

### Subjects

We used three rhesus macaques (*Macaca mulatta*), two male and one female, to study behavior on an asymmetric-reward response-time random-dot motion discrimination task (*Figure 1B*, see below). Prior to this study, monkeys F and C had been trained extensively on the equal-reward RT version of the task (*Ding and Gold, 2010*; *Ding and Gold, 2012b*; *Ding and Gold, 2012a*). Monkey A had been trained extensively on non-RT dots tasks (*Connolly et al., 2009*; *Bennur and Gold, 2011*), followed by >130 sessions of training on the equal-reward RT dots task. All training and experimental procedures were in accordance with the National Institutes of Health Guide for the Care and Use of Laboratory Animals and were approved by the University of Pennsylvania Institutional Animal Care and Use Committee (#804726).

### Behavioral task

Our task (*Figure 1B*) was based on the widely used random-dot motion discrimination task that typically has symmetric rewards (*Roitman and Shadlen, 2002*; *Ding and Gold, 2010*). Briefly, a trial started with presentation of a fixation point at the center of a computer screen in front of a monkey. Two choice targets appeared 0.5 s after the monkey acquired fixation. After a delay, the fixation point was dimmed and a random-dot kinematogram (speed: 6 °/s) was shown in a 5° aperture centered on the fixation point. For monkeys F and C, the delay duration was drawn from a truncated exponential distribution with mean = 0.7 s, max = 2.5 s, min = 0.4 s. For monkey A, the delay was set as 0.75 s. The monkey was required to report the perceived global motion direction by making a saccade to the corresponding choice target at a self-determined time (a 50 ms minimum latency was imposed to discourage fast guesses). The stimulus was immediately turned off when the monkeys' gaze left the fixation window (4°, 4°, and 3° square windows for monkey F, C, and A, respectively). Correct choices (i.e., saccades to the target congruent with actual motion direction) were rewarded with juice. Error choices were not rewarded and instead penalized with a timeout before the next trial began (timeout duration: 3 s, 0.5–2 s, and 2.5 s, for monkeys F, C, and A, respectively).

On each trial, the motion direction was randomly selected toward one of the choice targets along the horizontal axis. The motion strength of the kinematogram was controlled as the fraction of dots moving coherently to one direction (coherence). On each trial, coherence was randomly selected from 0.032, 0.064, 0.128, 0.256, and 0.512 for monkeys F and C, and from 0.128, 0.256, 0.512, and 0.75 for monkey A. In a subset of sessions, coherence levels of 0.064, 0.09, 0.35, and/or 0.6 were also used for monkey A.

We imposed two types of reward context on the basic task. For the 'LR-Left' reward context, correct leftward saccades were rewarded with a larger amount of juice than correct rightward saccades. For the 'LR-Right' reward context, correct leftward saccades were rewarded with a smaller amount of juice than correct rightward saccades. The large:small reward ratio was on average 1.34, 1.91, and 2.45 for monkeys F, C, and A, respectively. Reward context was alternated between blocks and constant within a block. Block changes were signaled to the monkey with an inter-block interval of 5 s. The reward context for the current block was signaled to the monkey in two ways: 1) in the first trial after a block change, the two choice targets were presented in blue and green colors, for small

and large rewards, respectively (this trial was not included for analysis); and 2) only the highest coherence level (near 100% accuracy) was used for the first two trials after a block change to ensure that the monkey physically experienced the difference in reward outcome for the two choices. For the rest of the block, choice targets were presented in the same color and motion directions and coherence levels were randomly interleaved.

We only included sessions in which there are more than 200 trials, more than eight coherences and more than eight trials for each coherence, motion direction and reward context (61, 37, and 43 sessions for monkey F, C, and A, respectively).

## Basic characterization of behavioral performance

Eye position was monitored using a video-based system (ASL) sampled at 240 Hz. RT was measured as the time from stimulus onset to saccade onset, the latter identified offline with respect to velocity (>40°/s) and acceleration (>8000°/s$^2$). Performance was quantified with psychometric and chronometric functions (*Figure 2* and *Figure 3*), which describe the relationship of motion strength (signed coherence, *Coh*, which was the proportion of the dots moving in the same direction, positive for rightward motion, negative for leftward motion) with choice and RT, respectively. Psychometric functions were fitted to a logistic function (*Equation 1*), in which $\lambda$ is the error rate, or lapse rate, independent of the motion information; $\alpha_0$ and ($\alpha_0 + \alpha_{rew}$) are the bias terms, which measures the coherence at which the performance was at chance level, in the LR-Right and LR-Left reward contexts, respectively. $\beta_0$ and ($\beta_0 + \beta_{rew}$) are the perceptual sensitivities in the LR-Right and LR-Left reward contexts, respectively.

$$P_{rightwardchoice} = \lambda + (1 - 2\lambda) \times \frac{1}{1 + e^{-Sensitivity(Coh - Bias)}}$$

(1)

## Reward-biased drift-diffusion model

To infer the computational strategies employed by the monkeys, we adopted the widely used accumulation-to-bound framework, the drift-diffusion model (DDM; *Figure 1A*). In the standard DDM, motion evidence is modeled as a random variable following a Gaussian distribution with a mean linearly proportional to the signed coherence and a fixed variance. The decision variable (DV) is modeled as temporal accumulation (integral) of the evidence, drifting between two decision bounds. Once the DV crosses a bound, evidence accumulation is terminated, the identity of the decision is determined by which bound is crossed, and the decision time is determined by the accumulation time. RT is modeled as the sum of decision time and saccade-specific non-decision times, the latter accounting for the contributions of evidence-independent sensory and motor processes.

To model the observed influences of motion stimulus and reward context on monkeys' choice and RT behavior, we introduced two reward context-dependent terms: *z* specifies the relative bound heights for the two choices and *me* specifies the equivalent momentary evidence that is added to the motion evidence at each accumulating step. Thus, for each reward context, six parameters were used to specify the decision performance: *a*: total bound height; *k*: proportional scaling factor converting evidence to the drift rate; $t_0$ and $t_1$: non-decision times for leftward and rightward choices, respectively; and *z* and *me*. Similar approaches have been used in studies of human and animal decision making under unequal payoff structure and/or prior probabilities (*Voss et al., 2004*; *Bogacz et al., 2006*; *Diederich and Busemeyer, 2006*; *Summerfield and Koechlin, 2010*; *Hanks et al., 2011*; *Mulder et al., 2012*).

To fit the monkeys' data, we implemented hierarchical DDM fitting using an open-source package in Python, which performs Bayesian estimates of DDM parameters based on single-trial RTs (*Wiecki et al., 2013*). This method assumes that parameters from individual sessions are samples from a group distribution. The initial prior distribution of a given parameter is determined from previous reports of human perceptual performance and is generally consistent with monkey performance on equal reward motion discrimination tasks (*Ding and Gold, 2010*; *Matzke and Wagenmakers, 2009*). The posterior distributions of the session- and group-level parameters are estimated with Markov chain Monte Carlo sampling. The HDDM was fit to each monkey separately.

For each dataset, we performed 5 chains of sampling with a minimum of 10000 total samples (range: 10000 – 20000; burn-in: 5000 samples) and inspected the trace, autocorrelation and marginal posterior histogram of the group-level parameters to detect signs of poor convergence. To ensure

similar level of convergence across models, we computed the Gelman-Rubin statistic (R-hat) and only accepted fits with R-hat <1.01.

To assess whether reward context modulation of both *z* and *me* was necessary to account for monkeys' behavioral data, we compared fitting performance between the model with both terms ('full') and reduced models with only one term ('z-only' and 'me-only'). Model selection was based on the deviance information criterion (DIC), with a smaller DIC value indicating a preferred model. Because DIC tends to favor more complex models, we bootstrapped the expected ΔDIC values, assuming the reduced models were the ground truth, using trial-matched simulations. For each session, we generated simulated data using the DDM, with single-session parameters fitted by *me*-only or *z*-only HDDM models and with the number of trials for each direction × coherence × reward context combination matched to the monkey's data for that session. These simulated data were then re-fitted by all three models to estimate the predicted ΔDIC, assuming the reduced model as the generative model.

To test an alternative model, we also fitted monkeys' data to a DDM with collapsing bounds (*Zylberberg et al., 2016*). This DDM was constructed as the expected first-stopping-time distribution given a set of parameters, using the PyMC module (version 2.3.6) in Python (version 3.5.2). The three model variants, 'full', '*me*-only' and '*z*-only', and their associated parameters were the same as in HDDM, except that the total bound distance decreases with time. The distance between the two choice bounds was set as

$$a \left/ \left(1 + e^{\beta(t-d)}\right)\right.$$

where *a* is the initial bound distance, *β* determines the rate of collapsing, and *d* determines the onset of the collapse. Fitting was performed by computing the maximum *a posteriori* estimates, followed by Markov chain Monte Carlo sampling, of DDM parameters given the experimental RT data.

## Sequential analysis

To examine possible sequential choice effects, for each monkey and session we fitted the choice data to three logistic functions. Each function was in the same form as *Equation 1* but with one of four possible additional terms describing a sequential effect based on whether the previous trial was correct or not, and whether the previous trial was to the large or small reward target. The sequential effect was assessed via a likelihood-ratio test for $H_0$: the sequential term in *Equation 2* = 0, p<0.05

$$P_{rightwardchoice} = \lambda + (1 - 2\lambda) \times \frac{1}{1 + e^{-Sensitivity(Coh-(Bias+Bias_{seq}))}} \tag{2}$$

$Bias_{seq}$ was determined using indicator variables for the given sequential effect and the reward context (e.g., LR-Right context, previous correct LR choice): $I_{seq} \times I_{rew} \times \boldsymbol{\alpha}_{seq}$, where $I_{rew} = \pm 1$ for LR-Right/LR Left reward contexts.

$I_{seq} = I_{prevLR-prevCorrect}, I_{prevLR-prevError}, I_{prevSR-prevCorrect},$ and $I_{prevSR-prevError}$ for the 4 types of sequential effects (note that there were not enough trials to compute previous error SR choice).

## Optimality analysis

To examine the level of optimality of the monkeys' performance, we focused on two reward functions: reward rate (RR, defined as the average reward per second) and reward per trial (RTrial, defined as the average reward per trial) for a given reward context for each session. To estimate the reward functions in relation to *me* and *z* adjustments for a given reward context, we numerically obtained choice and RT values for different combinations of *z* (ranging from 0 to 1) and *me* (ranging from −0.6 to 0.6 coherence unless otherwise specified), given *a*, *k*, and non-decision time values fitted by the full model. We then calculated RR and RTrial, using trial-matched parameters, including the actual ITI, timeout, and large:small reward ratio. $RR_{max}$ and $RTrial_{max}$ were identified as the maximal values given the sampled *me-z* combinations, using 1000 trials for each coherence ×direction condition. Optimal *me* and *z* adjustments were defined as the *me* and *z* values corresponding to $RR_{max}$ or $RTrial_{max}$. $RR_{predict}$ and $RTrial_{predict}$ were calculated with the fitted *me* and *z* values in the full model.

## Acknowledgments

We thank Takahiro Doi for helpful comments, Javier Caballero and Rachel Gates for animal training, Jean Zweigle for animal care, and Michael Yoder for data entry.

## Additional information

### Competing interests

Joshua I Gold: Reviewing editor, *eLife*. The other authors declare that no competing interests exist.

### Funding

| Funder | Grant reference number | Author |
| --- | --- | --- |
| National Eye Institute | R01-EY022411 | Joshua I Gold Long Ding |
| University of Pennsylvania | University Research Foundation Pilot Award | Long Ding |
| Hearst Foundations | Graduate student fellowship | Yunshu Fan |

The funders had no role in study design, data collection and interpretation, or the decision to submit the work for publication.

### Author contributions

Yunshu Fan, Data curation, Formal analysis, Validation, Investigation, Visualization, Methodology, Writing—original draft, Writing—review and editing; Joshua I Gold, Conceptualization, Resources, Software, Formal analysis, Supervision, Funding acquisition, Visualization, Methodology, Writing—review and editing; Long Ding, Conceptualization, Resources, Data curation, Software, Formal analysis, Supervision, Funding acquisition, Validation, Investigation, Visualization, Methodology, Writing—original draft, Project administration, Writing—review and editing

### Author ORCIDs

Yunshu Fan  http://orcid.org/0000-0003-2597-5173
Joshua I Gold  http://orcid.org/0000-0002-6018-0483
Long Ding  http://orcid.org/0000-0002-1716-3848

### Ethics

Animal experimentation: All training and experimental procedures were in accordance with the National Institutes of Health Guide for the Care and Use of Laboratory Animals and were approved by the University of Pennsylvania Institutional Animal Care and Use Committee (#804726).

### Decision letter and Author response

Decision letter https://doi.org/10.7554/eLife.36018.sa1
Author response https://doi.org/10.7554/eLife.36018.sa2

## Additional files

### Supplementary files

• Transparent reporting form

### Data availability

Raw data used during this study are included as the supporting files.

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
