## [Decision Letter]

Thank you for submitting your article "Ongoing, rational calibration of reward-driven perceptual biases" for consideration by *eLife*. Your article has been reviewed by three peer reviewers, including Peter Latham as the Reviewing Editor, and the evaluation has been overseen by Richard Ivry as the Senior Editor. The following individual involved in review of your submission has agreed to reveal his identity: Roger Ratcliff (Reviewer #3).

The reviewers have discussed the reviews with one another and the Reviewing Editor has drafted this decision to help you prepare a revised submission.

Summary:

The authors consider a relatively standard 2AFC task in which monkeys view a random dot kinematogram and have to decide whether the dots are moving to the right or left. To make things slightly more interesting than usual, the rewards are asymmetric: in blocks of 30-50 trials, one saccade directions receives higher reward than the other.

The monkey learned to take the asymmetry into account: the more uncertain they were, the more they favored the direction that had higher reward. However, they were slightly suboptimal: they received about 97% of the reward they could have received if they had used an optimal policy.

To explain the suboptimality, the authors used a "satisficing" gradient-based learning rule. Essentially, the monkeys follow the gradient in trial averaged reward rate until they achieve 97% performance, and then stop learning. For such a model, the final values of the parameters depend on initial conditions. The authors found, however, that, under their model, all the animals used the same initial condition; so-called "over-me".

Essential revisions:

There were a lot of things we liked about this paper (more on that below). However, all of us had problems with the "satisficing" learning rule. We know that's an essential feature of the paper, but it seems like a near untenable hypothesis. How, for instance, can the animal know when it reaches 97%? Presumably there's nothing special about 97%, but that still leaves the problem: how can the animal know when it's near the optimum, and so turn off learning? For the satisficing learning rule to be a viable explanation, these questions need to be answered.

Note that there are reasons to turn down the learning rate; see, for example, Aitchison et al., 2017, and Kirkpatrick et al., PNAS (2016), 106:10296--10301. But those approaches – which essentially turn down the learning rate when the synapses become more sure of their true values – would put a slightly different spin on the paper. In addition, the authors need to try other explanations. For instance, because learning is stochastic, the animal never reaches an optimum. And if the energy surface has a non-quadratic maximum (as it appears the energy surface does in this case), there will be bias. If the bias matches the observed bias, that would be a strong contender for a viable model. It's also possible that the model class used by the monkeys does not contain the true model, which could happen if z and me are tied in some way. That seems like an unlikely model, but probably not less likely than a model that turns down the learning rate. It is, at the very least, worth mentioning.

In addition, you should look for sequential effects: does behavior on one trial depend on the outcome of the previous trial? If so, can you link these to changes in me and z? If so, that could shed light on which model, if any, is correct. And it would be a nice addition to the analysis so far, which is primarily steady state.

On the plus side, the paper is short, well-written, and to the point, and the findings are novel and interesting. We're highly sympathetic to the idea that animals adopt satisficing solutions as opposed to optimal ones in many settings. We also think the ability to account for idiosyncratic differences in performance of different animals is a very nice result.

[Editors' note: further revisions were requested prior to acceptance, as described below.]

Thank you for resubmitting your work entitled "Ongoing, rational calibration of reward-driven perceptual biases" for further consideration at *eLife*. Your revised article has been reviewed by three reviewers, one of whom is a member of our Board of Reviewing Editors, and the evaluation has been overseen by Richard Ivry as the Senior Editor.

The manuscript has been improved but there are some remaining issues that need to be addressed before acceptance, as outlined below:

We see no fundamental obstacles to acceptance. However, there are still some things that are not clear. And in one case it appears that we were not sufficiently clear, so there is still a small amount of work to be done. Following are our suggestions; hopefully they will be crystal clear.

1) At least two of the reviewers were somewhat confused by the stopping rule. We think we eventually understood things, and the rule is in fact simple: stop learning when the reward reaches a certain (good enough) value. However, this is surprisingly hard to extract. We have a couple of suggestions to fix this:

a) You say:

"We next consider if and how a consistent, adaptive process used by all three monkeys could lead to these idiosyncratic and not-quite optimal patterns of adjustments."

This is not all that informative – basically, you're saying you have a model, but you're not going to tell the reader what it is. Why not say:

"We show that this can be explained by a model in which the monkeys are initially over-biased, adjust their model parameters to increase reward, but stop learning when the reward is high enough, but not maximum."

b) In Figure 8 legend, you say "All four searches stopped when the reward exceeded the monkeys' RTrial_predict_ in that session." This is hard to make sense of, for two reasons. First, we had to go back and figure out what RTrial_predict_ is. Second, the implication is that there's something special about RTrial_predict_. In fact, the point is that you stop integrating when the reward reaches the reward the monkeys got on that trial. It would be a lot easier to understand if you just said that, without mentioning RTrial_predict_.

c) Identical comments apply to the third paragraph of the subsection “The monkeys’ adaptive adjustments were consistent with a satisficing, gradient based learning process”.

2) The added paragraph was not so clear:

“Our last assumption was that the monkeys terminated adjustments as soon as they reached a good-enough reward outcome. […] Moreover, the updating process could use step sizes that are fixed or adaptive to the gradient of a reward-related cost function (Aitchison et al., 2017 and Kirkpatrick et al., 2017).”

The first two sentences are fine, but after that one would have to know those papers inside and out to understand what's going on. I think all you need to say is something like:

(Aitchison et al., 2017 and Kirkpatrick et al., 2017) proposed a model in which learning rates decreased as synapses become more certain. In this scheme, learning can become very slow near an optimum, and might account for our results.

3) In our previous review, we said

In addition, the authors need to try other explanations. For instance, because learning is stochastic, the animal never reaches an optimum. And if the energy surface has a non-quadratic maximum (as it appears the energy surface does in this case), there will be bias. If the bias matches the observed bias, that would be a strong contender for a viable model.

In your response, you seemed to be able to guess what would happen in this case. We admire you if you are indeed correct, but we think it will require simulations. In particular, you need to run a model of the form

Delta z = eta dR/dz + noise

Delta me = eta dR/dme + noise

where R is reward. The average reward value of z and me under this model will, for a non-quadratic surface, but slightly biased. We think it's important to determine, numerically, what that bias is. Given that you're set up to solve an ODE, it shouldn't be much work to check what happens to the above equations.

4) You should cite Ratcliff, 1985, – as far as we know, that was the first paper to have bias in drift rates and starting point as 2 possibilities.

5) Take out this quote:

"… they may not capture the substantial variability under different conditions and/or across individual subjects".

6) Add in the Green and Swets reference near Figure 2 and note that signal detection theory does not have a model of criterion setting and does not achieve optimal performance where that has been studied.

---

## [Author Response]

Essential revisions:There were a lot of things we liked about this paper (more on that below). However, all of us had problems with the "satisficing" learning rule. We know that's an essential feature of the paper, but it seems like a near untenable hypothesis. How, for instance, can the animal know when it reaches 97%? Presumably there's nothing special about 97%, but that still leaves the problem: how can the animal know when it's near the optimum, and so turn off learning? For the satisficing learning rule to be a viable explanation, these questions need to be answered.

We apologize for the confusion. We agree with the reviewers that the monkeys likely did not know when they reached 97% of the maximum reward per trial or reward rate, which would imply that the monkeys also knew the location of the peak of the reward surface. In fact, the model simulations that we used to compare to the monkeys’ data did not use that stopping criterion. Instead, we assumed that the monkeys used an absolute reward criterion, computed as the expected reward based on the session-specific task and model parameters (using RTrial for Figure 8 and RR for Figure 8—figure supplement 1). For example, they might be “satisfied” with 3 ml juice per 10 trials for reward per trial, or 3 ml juice per minute for reward rate, to turn off learning.

We used a 97% maximum reward per trial or reward rate as the stopping point only for the illustrations of gradient trajectories in Figure 7, Figure 7—figure supplement 1, Figure 8—figure supplement 3 and 4. Note that in these simulations, the distinction between a “97% of maximum” versus an “absolute reward” criterion is arbitrary and depends only on our choice of model parameters.

We have revised the text to highlight these points, as follows:

Figure 8 legend: “All four searches stopped when the reward exceeded the monkeys’ RTrial_predict_ in that session.”

Results: “The lines on the RTrial heatmap represent the trajectories of a gradient-tracking procedure that adjusts *me* and *z* values to increase RTrial until a termination point (for illustration, here we used 97% of the maximum possible value).”

Note that there are reasons to turn down the learning rate; see, for example, Aitchison, Pouget, and Latham, 2017, and Kirkpatrick et al., PNAS (2016), 106:10296--10301. But those approaches – which essentially turn down the learning rate when the synapses become more sure of their true values – would put a slightly different spin on the paper.

Thank you for pointing us to these relevant studies. We have added a paragraph in the Discussion to speculate on different mechanisms of stopping:

“Our last assumption was that the monkeys terminated adjustments as soon as they reached a good-enough reward outcome. […] Moreover, the updating process could use step sizes that are fixed or adaptive to the gradient of a reward-related cost function (Aitchison et al., 2017 and Kirkpatrick et al., 2017)”

To explore these points further, we simulated an updating process with a stopping rule that was based on the gradient of the reward function. As shown in Author response image 1, in the same example session as in Figure 8, the reward and gradient-based updating and termination rules landed at similar end points. However, in general the reward-based updating process that we used in the manuscript approximated the monkeys’ data better (Author response image 1). We have not included this analysis in the revised manuscript but would be happy to do so at the reviewers’ discretion.

**Author response image 1. respfig1:** Comparison between reward and gradient-based updating process. (**A**) example over-*me* reward (magenta) and gradient (cyan)-based updating process The same example as shown in Figure 8A. Gray circle indicates monkeys’ *me* and *z*. (**B**) Same format as Figure 8B. Scatterplot of end points for the two updating processes.

In addition, the authors need to try other explanations. For instance, because learning is stochastic, the animal never reaches an optimum. And if the energy surface has a non-quadratic maximum (as it appears the energy surface does in this case), there will be bias. If the bias matches the observed bias, that would be a strong contender for a viable model.

The reviewers are correct that the energy surface has a non-quadratic maximum and the gradient trajectories across the *me/z* space show certain biases (Figure 7 and Figure 7—figure supplement 1). For example, the area covered by the magenta or blue gradient lines is larger than the area covered by the green lines, indicating that if a monkey randomly picks a starting point within the *me/z* space, it is more likely to end up with one of two combinations: either an overly biased *me* and a non-adaptive *z* or an overly biased *z* and a non-adaptive *me*. However, these two combinations were not equally represented in our data. Instead, all three monkeys we tested showed a strong bias toward the first. Therefore, we do not think that the inherent bias in the energy surface alone can explain the observed patterns of behavior. As we now note in the text: “Figure 7 also illustrates why assumptions about the starting point of this adaptive process are important: randomly selected starting points would result in learned *me* and *z* values distributed around the peak of the reward function, whereas the data (e.g., Figure 5C) show distinct clustering that implies particular patterns of starting points.”

It's also possible that the model class used by the monkeys does not contain the true model, which could happen if z and me are tied in some way. That seems like an unlikely model, but probably not less likely than a model that turns down the learning rate. It is, at the very least, worth mentioning.

Thank you for this suggestion. We agree that this possibility is important and should be acknowledged. We have added a sentence in Discussion (subsection “Considerations for assessing optimality and rationality”): “The particular form of DDM that we used produced reasonably good, but not perfect, fits to the monkeys’ data. These results support the utility of the DDM framework but also underscore the fact that we do not yet know the true model, which could impact our optimality assessment.”

In addition, you should look for sequential effects: does behavior on one trial depend on the outcome of the previous trial? If so, can you link these to changes in me and z? If so, that could shed light on which model, if any, is correct. And it would be a nice addition to the analysis so far, which is primarily steady state.

Thank you for this suggestion. We did these analyses and found no evidence for consistent or substantial sequential choice effects for the three monkeys. We therefore conclude that these effects do not substantially impact our results. We now describe these findings in Results (subsection “The monkeys’ biases reflected changes in reward context and perceptual sensitivity”, first paragraph and Figure 2—figure supplement 1).

We also added a paragraph in the Materials and methods section describing the sequential analysis (subsection “Sequential analysis”).

On the plus side, the paper is short, well-written, and to the point, and the findings are novel and interesting. We're highly sympathetic to the idea that animals adopt satisficing solutions as opposed to optimal ones in many settings. We also think the ability to account for idiosyncratic differences in performance of different animals is a very nice result.

Thank you for these encouraging comments!

[Editors' note: further revisions were requested prior to acceptance, as described below.]

The manuscript has been improved but there are some remaining issues that need to be addressed before acceptance, as outlined below:We see no fundamental obstacles to acceptance. However, there are still some things that are not clear. And in one case it appears that we were not sufficiently clear, so there is still a small amount of work to be done. Following are our suggestions; hopefully they will be crystal clear.1) At least two of the reviewers were somewhat confused by the stopping rule. We think we eventually understood things, and the rule is in fact simple: stop learning when the reward reaches a certain (good enough) value. However, this is surprisingly hard to extract. We have a couple of suggestions to fix this:a) You say:"We next consider if and how a consistent, adaptive process used by all three monkeys could lead to these idiosyncratic and not-quite optimal patterns of adjustments."This is not all that informative – basically, you're saying you have a model, but you're not going to tell the reader what it is. Why not say:"We show that this can be explained by a model in which the monkeys are initially over-biased, adjust their model parameters to increase reward, but stop learning when the reward is high enough, but not maximum."

Thank you for the suggestion. We have incorporated this way of phrasing. The new text is: “We next show that these shifts can be explained by a model in which the monkeys are initially over-biased, then adjust their model parameters to increase reward and stop learning when the reward is high enough, but not at its maximum possible value.”

b) In Figure 8 legend, you say "All four searches stopped when the reward exceeded the monkeys' RTrial_predict_ in that session." This is hard to make sense of, for two reasons. First, we had to go back and figure out what RTrial_predict_ is. Second, the implication is that there's something special about RTrial_predict_. In fact, the point is that you stop integrating when the reward reaches the reward the monkeys got on that trial. It would be a lot easier to understand if you just said that, without mentioning RTrial_predict_.

Thank you for pointing this out. However, the stopping rule we used was not when the reward reaches the reward the monkeys got on that trial, but rather when the reward reaches the average reward the monkeys got on in that session. To make the sentence easier to understand, we changed it into “All four searches stopped when the reward exceeded the average reward the monkey received in that session (RTrial_predict_), estimated from the corresponding best-fitting model parameters and task conditions”.

c) Identical comments apply to the third paragraph of the subsection “The monkeys’ adaptive adjustments were consistent with a satisficing, gradient based learning process”.

We changed “RTrial_predict_” in that sentence into “average reward in that session (RTrial_predict_)”.

2) The added paragraph was not so clear:“Our last assumption was that the monkeys terminated adjustments as soon as they reached a good-enough reward outcome. […] Moreover, the updating process could use step sizes that are fixed or adaptive to the gradient of a reward-related cost function (Aitchison et al., 2017 and Kirkpatrick et al., 2017).”The first two sentences are fine, but after that one would have to know those papers inside and out to understand what's going on. I think all you need to say is something like:(Aitchison et al., 2017 and Kirkpatrick et al., 2017) proposed a model in which learning rates decreased as synapses become more certain. In this scheme, learning can become very slow near an optimum, and might account for our results.

Thank you for the suggestion. We have changed the last two sentences to: “For example, the learning rate for synaptic weights might decrease as the presynaptic and postsynaptic activities become less variable (Aitchison et al., 2017; Kirkpatrick et al., 2017). In this scheme, learning gradually slows down as the monkey approaches the plateau on the reward surface, which might account for our results.”

3) In our previous review, we saidIn addition, the authors need to try other explanations. For instance, because learning is stochastic, the animal never reaches an optimum. And if the energy surface has a non-quadratic maximum (as it appears the energy surface does in this case), there will be bias. If the bias matches the observed bias, that would be a strong contender for a viable model.In your response, you seemed to be able to guess what would happen in this case. We admire you if you are indeed correct, but we think it will require simulations. In particular, you need to run a model of the formDelta z = eta dR/dz + noiseDelta me = eta dR/dme + noisewhere R is reward. The average reward value of z and me under this model will, for a non-quadratic surface, but slightly biased. We think it's important to determine, numerically, what that bias is. Given that you're set up to solve an ODE, it shouldn't be much work to check what happens to the above equations.

Thank you for the suggestion. We simulated this model at random starting points with different scaling factors (eta) and different noise levels. As shown in Author response image 2, when the reward-gradient updating processes start from random locations on the reward function, the end points tend to cover all directions relative to the peak, with majority of the end points clustered in the over-me, under-z region and the over-z, under-me region. This result is inconsistent with our monkeys’ behaviors, which deviated from the peak only in the over-*me*, under-*z* region. Therefore we do not think this model can explain our data. We have not included this analysis in the revised manuscript but would be happy to do so at your discretion.

**Author response image 2. respfig2:** Gradient updating with randomness in the starting location and each updating step does not generate the biased end-point pattern seen in the data. (**A**) Simulation of reward-gradient updating trajectories starting from random locations (white circles) on the reward function. Each updating step = scaling factor x gradient + noise (noise along the *me* and *z* dimensions were generated independently from the same Gaussian distribution). The updating process stopped when reward exceeded 97% of the maximum (red circles indicate end points). Note that the end-points of the updating process were located all around the reward-function plateau. The reward function was from the LR-R blocks in an example session of monkey C. (**B**) Polar histograms of the angle of the end-points relative to the peak of the reward function, showing end-points scattered all around the peak with clusters in two locations.

4) You should cite Ratcliff, 1985, – as far as we know, that was the first paper to have bias in drift rates and starting point as 2 possibilities.

Thank you for pointing out this reference. We have added this citation in the second paragraph of the Introduction.

5) Take out this quote:"… they may not capture the substantial variability under different conditions and/or across individual subjects".

Thank you for the suggestion. However, we have opted to keep this sentence, which we think makes a useful and accurate point.

6) Add in the Green and Swets reference near Figure 2 and note that signal detection theory does not have a model of criterion setting and does not achieve optimal performance where that has been studied.

Given that Figure 2 is about logistic regression, not signal detection theory, we have decided not to include this reference here.